# Stochastically Dominant Peer Prediction

**Yichi Zhang**
DIMACS, Rutgers University
yz1636@dimacs.rutgers.edu

**Shengwei Xu**
University of Michigan, Ann Arbor
shengwei@umich.edu

**David Pennock**
DIMACS, Rutgers University
dpennock@dimacs.rutgers.edu

**Grant Schoenebeck**
University of Michigan, Ann Arbor
schoeneb@umich.edu

## Abstract

Eliciting reliable human feedback is essential for many machine learning tasks, such as learning from noisy labels and aligning AI systems with human preferences. Peer prediction mechanisms incentivize truthful reporting without ground truth verification by scoring agents based on correlations with peers. Traditional mechanisms, which ensure that truth-telling maximizes the **expected scores** in equilibrium, can elicit honest information while assuming agents' utilities are **linear functions** of their scores. However, in practice, non-linear payment rules are usually preferred, or agents' utilities are inherently non-linear.

We propose *stochastically dominant truthfulness (SD-truthfulness)* as a stronger guarantee: the score distribution of truth-telling stochastically dominates all other strategies, incentivizing truthful reporting for a wide range of monotone utility functions. Our first observation is that no existing peer prediction mechanism naturally satisfies this criterion without strong assumptions. A simple solution—rounding scores into binary lotteries—can enforce SD-truthfulness, but often degrades *sensitivity*, a key property related to fairness and statistical efficiency. We demonstrate how a more careful application of rounding can better preserve sensitivity. Furthermore, we introduce a new enforced agreement (EA) mechanism that is theoretically guaranteed to be SD-truthful in binary-signal settings and, under mild assumptions, empirically achieves the highest sensitivity among all known SD-truthful mechanisms.

## 1 Introduction

Information elicitation studies how to design mechanisms that incentivize honest and high-effort human feedback. In this framework, a principal seeks reliable responses regarding a set of tasks from strategic agents who are motivated by the designed rewards. For example, AI companies collect human preferences to align language models via RLHF [Ouyang et al., 2022], and MOOCs use peer grading to evaluate complex assignments at scale Piech et al. [2013]. A key challenge is designing scoring mechanisms that accurately assess the informativeness of agents' reports.

Peer prediction provides an elegant solution for settings where ground truth is unavailable or costly to obtain, such as in subjective tasks [Miller et al., 2005]. Instead of relying on external validation, peer prediction scores agents based on the correlation between their reports and those of their peers. The scoring mechanism is designed to be *truthful* so that everyone reporting truthfully forms a Bayesian Nash equilibrium, ensuring that truth-telling maximizes each agent's **expected score**.

A key assumption underlying the effectiveness of existing peer prediction mechanisms is that agents' utility is a **linear function** of their score. This assumption is reasonable in some cases, such as when agents are solely motivated by monetary payments, which can be straightforwardly designed as a

39th Conference on Neural Information Processing Systems (NeurIPS 2025).

linear transformation of the peer prediction scores. However, in practice, non-linear payment rules such as threshold bonuses that reward workers only when performance exceeds a cutoff are often preferred for their budget efficiency and flexibility [Zhang and Schoenebeck, 2023a]. Moreover, agents' intrinsic utility functions are sometimes inherently non-linear. For instance, student graders care more about their final letter grades (e.g., A, B, C) than the real-valued numerical scores assigned by the mechanism. When agents' utilities are non-linear functions of their scores, truth-telling is no longer guaranteed to be an equilibrium strategy.

Motivated by this limitation, we introduce the concept of *stochastic dominance-truthful (SD-truthful)* peer prediction mechanisms. This stronger notion of truthfulness requires that an agent's **score distribution**, when everyone reports truthfully, first-order stochastically dominates the score distribution under any unilateral deviation. As a result, SD-truthful mechanisms incentivize truth-telling for agents with **any utility function that is monotone increasing with the score**.

## 1.1 Our Results

Our first observation is that **no existing peer prediction mechanism is naturally SD-truthful under general information structures**, i.e., under arbitrary correlations between agents' signals. We examine three widely studied mechanisms that compute scores as the average of multiple simple discrete scores, including *Output Agreement (OA)*, *Peer Truth Serum (PTS)* [Faltings et al., 2017], and *Correlated Agreement (CA)* [Shnayder et al., 2016]. Mechanisms with complex scoring rules—such as those based on continuous-valued estimates of mutual information Kong and Schoenebeck [2019], Kong [2020], Schoenebeck and Yu [2020]—are unlikely to satisfy SD-truthfulness. Yet even these simple mechanisms fail to guarantee SD-truthfulness without strong assumptions on the underlying information structure.

Our second observation is that **achieving SD-truthfulness is easy in principle but challenging to do well in practice**. Given any truthful mechanism $\mathcal{M}$ with bounded scores $S^{\mathcal{M}} \in [S_{\inf}, S_{\sup}]$, we can define a probability by normalizing the score: $\lambda^{\mathcal{M}} = \frac{S^{\mathcal{M}} - S_{\inf}}{S_{\sup} - S_{\inf}} \in [0, 1]$. We then score agents via a binary lottery: $\tilde{S}^{\mathcal{M}} = 1$ with probability $\lambda^{\mathcal{M}}$ and 0 otherwise. Although this approach, called the *direct rounding reduction*, trivially satisfies SD-truthfulness, it greatly reduces *sensitivity*—an important property related to the efficiency and ex-post fairness of a mechanism. At a high level, sensitivity measures how much a mechanism's score responds to changes in the information contained in an agent's reports, normalized by the standard deviation of the scores.

To overcome this limitation, **we propose the *partition rounding reduction*, which helps partially recover the sensitivity of the original mechanism**. The idea is to divide the questions into $K$ disjoint subsets and apply the original mechanism separately within each subset to compute an individual score. The final score for an agent is then the average of $K$ i.i.d. individual scores after applying direct rounding. Intuitively, this approach improves sensitivity by reducing variance through the aggregation of multiple independent scores.

However, the partition rounding process cannot fully recover the sensitivity of the original mechanisms, implying additional room for more sensitive peer prediction mechanisms. **We introduce a novel mechanism called *enforced agreement (EA)* that is shown to be the most sensitive SD-truthful mechanism in the binary-signal setting**. EA aims to replicate the key advantage of OA—its high sensitivity—while relaxing the strong assumptions on the information structure required for SD-truthfulness.

EA works by enforcing a pre-determined empirical distribution of signals. In the binary setting, if Alice reports 0 on $m_0 > n_0$ questions (where $n_0$ is the enforced count), the mechanism randomly flips $m_0 - n_0$ of them to 1. This trick removes agents' incentive to over-report majority signals, a strategy that undermines the truthful properties of OA. However, EA is only truthful but not SD-truthful when signals are non-binary, highlighting an important future challenge in designing high-sensitivity SD-truthful mechanisms for more complex settings.

Lastly, our empirical results reinforce and complement our theoretical insights. In the binary-signal setting, we show that even a prior-free implementation of EA—simply enforcing a uniform distribution—outperforms other SD-truthful mechanisms in most settings, including those that require prior knowledge. For non-binary-signal settings, the partition rounding reduction of PTS emerges as the most sensitive SD-truthful mechanism.

Table 1 summarizes a comparison of SD-truthful (implementations of) peer prediction mechanisms with respect to the assumptions they require and their sensitivities.[1]

| Mechanisms | Requirements for SD-truthfulness |
|---|---|
| Output Agreement (OA) | Self-dominating signal |
| Enforced Agreement (EA) [This paper] | Binary and self-predicting signal |
| Peer Truth Serum (PTS) [Faltings et al., 2017] | Self-predicting signal |
| Correlated Agreement (CA)[Shnayder et al., 2016] | Any signal |
| Matching Agreement (MA) [Zhang and Schoenebeck, 2023b] | Any signal |

Table 1: A Comparison of Discussed Peer Prediction Mechanisms ordered by sensitivity (from high to low), where PTS, CA, and MA require the partition rounding reduction to achieve SD-truthfulness.

## 2  Related Work

The idea of peer prediction was proposed by Miller et al. [2005]. As the first discussion, their mechanism is not *detail-free*, i.e. the mechanism requires prior knowledge of the information structure (how agents' signals are correlated) to obtain truthfulness. There are two ideas to mitigate this limitation: (i) the single-task mechanism, which elicits the information structure directly from agents [Prelec, 2004]; and (ii) the multi-task mechanism, which assumes each agent responds to multiple i.i.d. tasks but only requires eliciting signals rather than predictions [Dasgupta and Ghosh, 2013].

Our study is primarily relevant to the multi-task setting. We categorize multi-task peer prediction mechanisms into two types: peer agreement mechanisms Dasgupta and Ghosh [2013], Faltings et al. [2017], Shnayder et al. [2016], Zhang and Schoenebeck [2023b], Agarwal et al. [2017] and mutual information mechanisms Kong and Schoenebeck [2019], Kong [2020], Schoenebeck and Yu [2020], Schoenebeck et al. [2021], Kong [2024]. In this paper, we primarily focus on peer agreement mechanisms, as SD-truthfulness requires the mechanism's score to take a relatively simple form. Intuitively, this is because a complex scoring mechanism provides more opportunities for a cheating strategy to obtain one of the highest scores with a larger probability than truth-telling, which breaks SD-truthfulness. In Appendix A, we provide a more detailed discussion on why several classic mutual information mechanisms are not SD-truthful.

The score of a peer agreement mechanism can be viewed as an average of multiple individual scores, where each individual score takes values from a small finite space, e.g. $\{0, 1\}$. For example, output agreement (OA) scores an agent based on the average number of questions that they agree with another agent. The first multi-task peer prediction mechanism lies in this category, which can achieve truthfulness in a binary-signal setting [Dasgupta and Ghosh, 2013]. The correlated agreement (CA) mechanism generalizes their idea to any finite-signal setting [Shnayder et al., 2016]. The peer truth serum (PTS) mechanism is a generalization of OA, which relaxes the self-dominating assumption to self-prediction. In addition, two follow-ups of CA aim to handle heterogeneous agents [Agarwal et al., 2017] and more general cheating strategies [Zhang and Schoenebeck, 2023b].

Moreover, we note that the idea of rounding peer prediction scores into a binary lottery (see Section 5) was previously proposed by Miller et al. [2005] to address risk-averse agents. What is new in our work is the comparative analysis of various rounding reductions based on sensitivity, allowing us to identify a more effective rounding approach. Furthermore, we formally prove that this idea extends beyond risk-averse agents and can be applied to any agents with increasing utility functions.

## 3  Model

We consider the multi-task peer prediction setting where two agents, Alice and Bob, answer the same $n$ i.i.d. questions. For each question $j$, the agents receive discrete signals $X_{a,j}, X_{b,j} \in \Sigma = \{0, \ldots, c-1\}$, drawn from a fixed joint distribution $\Pr(X_a, X_b)$. In cases with more than two agents, we can reduce to the case of two agents by, for any particular agent "Alice", choosing a random peer agent "Bob" for scoring.

---

[1]MA is an extension of CA, and therefore shares similar requirements and properties.

Each agent $i \in \{a, b\}$ applies a strategy $\theta_i \in \Theta$, which maps from signals to distributions over the same space $\Delta_\Sigma$. The report on question $j$ is denoted as $\hat{X}_{i,j} = \theta_i(X_{i,j})$. We highlight two types of strategies: the truth-telling strategy $\tau$, which always reports the received signal, and the uninformative strategy $\mu$, where reports are independent of the input. Examples of uninformative strategies include always reporting the same signal or randomly reporting signals according to the prior.

A peer prediction mechanism $\mathcal{M}$ maps vectors of reports $\hat{X}_a$ and $\hat{X}_b$ to a score for Alice. Given an information structure (i.e., the joint distribution over signals) and a mechanism, the score is a random function of strategies, denoted as $S^{\mathcal{M}}(\theta_a, \theta_b)$. The randomness arises from the signals, strategies, and the mechanism itself. When it is clear from context, we omit the superscript $\mathcal{M}$.

### 3.1 Truthful Guarantees

Existing peer prediction mechanisms focus on expected scores, guaranteeing that any deviation from truth-telling will only reduce the expected score. In this sense, truth-telling forms a Bayesian Nash Equilibrium. Since our analysis centers on equilibrium behavior, we assume Bob always tells the truth. Under this assumption, the score assigned by the mechanism reduces to a function of Alice's strategy $\theta$, written as $S^{\mathcal{M}}(\theta)$.

**Definition 3.1.** A peer prediction mechanism is truthful if $\mathbb{E}[S(\tau)] \geq \mathbb{E}[S(\theta)]$ for any $\theta \in \Theta$.

A truthful mechanism guarantees that if agents have linear utility in their scores, truth-telling is a Bayesian Nash Equilibrium—any deviation yields lower expected utility. However, as we argued earlier, assuming linear utility is often undesirable and, in some cases, unrealistic. Instead, we aim to design truthful mechanisms for a broader and more reasonable class of utility functions—those that are simply increasing in score. This motivation naturally leads to the concept of stochastic dominance.

**Definition 3.2.** A peer prediciton mechanism is stochastic dominance-truthful (SD-truthful) if the distribution of $S(\tau)$ first-order stochastic dominates (FOSD) the distribution of $S(\theta)$ for any $\theta \in \Theta$, i.e. $\Pr(S(\tau) \geq t) \geq \Pr(S(\theta) \geq t)$ for any $t \in \mathbb{R}$.

A standard result in microeconomic theory suggests that $S(\tau)$ FOSD $S(\theta)$ if and only if $\mathbb{E}[u(S(\tau))] \geq \mathbb{E}[u(S(\theta))]$ for every increasing utility function $u$ [Hadar and Russell, 1969]. Since linear utility functions are also increasing, every SD-truthful mechanism is also truthful.

### 3.2 Sensitivity

In practice, an effective scoring mechanism should meaningfully evaluate the information contained in agents' reports and reward more informative reports with higher scores. This motivates an additional design criterion, orthogonal to truthfulness, known as *sensitivity* [Zhang and Schoenebeck, 2023a].

To motivate the concept of sensitivity, we extend the model to incorporate agents' effort levels. This extension captures how the mechanism responds to marginal changes in information at equilibrium, i.e. how the score varies when all other agents exert full effort while one agent adjusts her effort. In particular, on each question, we assume Alice exerts effort with probability $e \in [0, 1]$, while Bob always exerts effort and reports truthfully. When Alice exerts effort, the signal pair $(X_a, X_b)$ is drawn from the joint distribution $\Pr(X_a, X_b)$; in the extreme case where Alice exerts no effort, the signals are independently drawn from the marginals $\Pr(X_a) \Pr(X_b)$. That is, exerting effort produces a signal $X_a$ that is correlated with $X_b$ according to the joint distribution, while shirking yields an uninformative guess drawn from the prior $\Pr(X_a)$. After observing the signal, Alice will report truthfully. Then, Alice's score under mechanism $\mathcal{M}$ becomes a function of $e$, denoted as $S^{\mathcal{M}}(e)$.

**Definition 3.3.** The sensitivity of a peer prediction mechanism $\mathcal{M}$ at effort level $e$ is $\delta^{\mathcal{M}}(e) := \frac{\nabla \mathbb{E}[S^{\mathcal{M}}(e)]}{\text{std}(S^{\mathcal{M}}(e))}$ where $\nabla$ denotes the derivative operator and std denotes the standard deviation.

Sensitivity measures how well a mechanism's expected score responds to changes in effort—and thus, changes in information. Consequently, a trivial mechanism that always assigns a score of 1, though truthful, has zero sensitivity, as its output is unaffected by effort.

Prior work [Zhang and Schoenebeck, 2023a] shows that when the score is normally distributed and agents are rewarded based on the ranking of scores, sensitivity is a sufficient statistic for *budgetary*

*efficiency*: mechanisms with higher sensitivity can elicit the same equilibrium effort level at a lower budgetary cost. Moreover, Xu et al. [2024] establish a one-to-one correspondence between sensitivity and *measurement integrity* [Burrell and Schoenebeck, 2023], implying that mechanisms with higher sensitivity generate scores more closely aligned with the true quality of responses. Together, these findings highlight sensitivity as a practically meaningful and theoretically grounded property of peer prediction mechanisms, motivating our choice to use it as an additional dimension for comparison.

**Our goal is to design SD-truthful mechanisms that also achieve high sensitivity.**

# 4 Prior Mechanisms

To the best of our knowledge, no existing peer prediction mechanism naturally achieves SD-truthfulness without imposing strong assumptions on the information structure. In this section, we introduce three exemplary multi-task peer prediction mechanisms and explain why they are not naturally SD-truthful to motivate our methods.

## 4.1 Output Agreement (OA)

The *output agreement (OA)* mechanism scores Alice based on the fraction of questions on which both agents agree. The final score of OA is the average of $n$ independent binary individual scores, i.e. $S^{OA} = \frac{1}{n} \sum_{i \in [n]} S_i^{OA}$ where $S_i^{OA} = 1$ if both agents' reports agree on question $i$ and 0 otherwise. However, OA encourages over-reporting of the majority signal, which is truthful only under a strong assumption.

**Definition 4.1.** Agents' signals are self-dominating if $\Pr(X_b = \sigma | X_a = \sigma) > \Pr(X_b = \sigma' | X_a = \sigma)$ for any $\sigma' \neq \sigma \in \Sigma$.

**Proposition 4.2.** *OA is truthful if agents' signals are self-dominating.*

In fact, OA is also SD-truthful when signals are self-dominating. This is because each individual score $S_i^{OA}$ takes only two values, meaning that $S_i^{OA}(\tau)$ FOSD $S_i^{OA}(\theta)$ if and only if $\Pr(S_i^{OA}(\tau) = 1) \geq \Pr(S_i^{OA}(\theta) = 1)$. Furthermore, the average of multiple i.i.d. scores that are each SD-truthful remains SD-truthful (see Section 5.2). Consequently, for OA, SD-truthfulness reduces to truthfulness.

## 4.2 Peer Truth Serum (PTS)

Self-dominance is usually considered a strong assumption. [Radanovic et al., 2016] propose the *Peer Truth Serum (PTS)* which incentivizes truth-telling under a weaker assumption.[2]

**Definition 4.3.** Agents' signals are self-predicting if $\Pr(X_b = \sigma | X_a = \sigma) > \Pr(X_b = \sigma | X_a = \sigma')$ for any $\sigma' \neq \sigma \in \Sigma$.

PTS assumes a symmetric, publicly known prior $R(\sigma) = \Pr(X_a = \sigma) = \Pr(X_b = \sigma)$. It repeatedly samples questions without replacement, and assigns an individual score $S_i^{PTS} = 1/R(\sigma)$ to Alice if $\hat{X}_{a,i} = \hat{X}_{b,i} = \sigma$, and 0 otherwise. The final score of PTS is the average of all $S_i^{PTS}$.

**Proposition 4.4** ([Faltings et al., 2017]). *PTS is truthful if agents' signals are self-predicting.*

**Is PTS SD-truthful?** The short answer is no. Each individual score $S_i^{PTS}$ takes at most $|\Sigma| + 1$ values, with the maximum being $S_{\max} = 1/R(\sigma_{\min})$, where $\sigma_{\min}$ is the least likely signal under the prior. A necessary condition for PTS to be SD-truthful is that truth-telling maximizes $\Pr\left(S_i^{PTS}(\tau) = S_{max}\right)$, which is the probability of Alice agreeing with Bob on signal $\sigma_{min}$. While truth-telling, this probability is the joint distribution $\Pr(X_a = \sigma_{min}, X_b = \sigma_{min})$. However, if Alice uses an uninformative strategy $\mu$ that always reports $\sigma_{min}$, this probability increases to $\Pr(X_b = \sigma_{\min})$. Therefore, although $\mu$ decreases the expected score, its distribution is not stochastically dominated by truth-telling, meaning that there exists an increasing utility function under which Alice strictly prefers $\mu$ to $\tau$.

---

[2]In the binary signal setting, self-prediction is equivalent to positive correlation, which is a strictly weaker condition than self-dominance.

### 4.3 Correlated Agreement (CA)

The correlated agreement (CA) mechanism [Shnayder et al., 2016] takes two questions to compute an individual score which has three possible values: -1, 0, and 1. Suppose the joint distribution $\Pr(X_a, X_b)$ is known. The CA mechanism first computes the delta matrix $\Delta_{\sigma,\sigma'} = \Pr(X_a = \sigma, X_b = \sigma') - \Pr(X_a = \sigma)\Pr(X_b = \sigma')$, which is the difference between the joint distribution and the product of marginal distributions. Let $T_\Delta(\sigma, \sigma') = \mathrm{Sign}(\Delta)_{\sigma,\sigma'}$ be the agreement function where $\mathrm{Sign}(x) = 1$ if $x > 0$ and 0 otherwise. Intuitively, $T_\Delta(\sigma, \sigma') = 1$ if the pair of signals $(\sigma, \sigma')$ appears more often on the same question than on distinct questions.

Then, CA repeatedly samples a bonus question $j$ and a penalty question $k$. For an individual score $S_i^{CA}$, Alice gets 1 if both agents agree on $j$ and gets $-1$ if her response on $j$ agrees with Bob's response on $k$, i.e. $S_i^{CA} = T_\Delta(\hat{X}_{a,j}, \hat{X}_{b,j}) - T_\Delta(\hat{X}_{a,j}, \hat{X}_{b,k})$. Since each bonus question can be paired with $n - 1$ penalties, the final score is the average over $n(n - 1)$ individual scores.

Intuitively, the CA mechanism works by encouraging correlations on the same question and penalizing correlations on distinct questions. Consequently, if Alice always reports the majority signal, the frequency of agreements on the bonus question will be identical to the frequency of agreements on the penalty questions, which results in a score of 0. Prior work shows that when $\Pr(X_a, X_b)$ is known (or accurately estimated), CA is truthful—and in fact, *informed truthful* [Shnayder et al., 2016].

**Is CA SD-truthful?** The answer depends on the implementation. The following example illustrates that the original implementation of CA described above is not SD-truthful.

*Example* 4.1. Suppose $n = 2$, so that Alice knows each of the two questions she answers will be used once as the bonus question and once as the penalty question. Her final score is the average of two individual scores, each with $S_i^{CA} \in \{-1, 0, 1\}$. Thus, the final score has five possible values: $S^{CA} \in \{-1, -0.5, 0, 0.5, 1\}$. A necessary condition for SD-truthfulness is that $\Pr(S^{CA}(\tau) = -1) \leq \Pr(S^{CA}(\theta) = -1)$ for any strategy $\theta$. However, we show that this does not hold in general.

Suppose the signals are positively correlated, i.e. $T_\Delta$ is a diagonal matrix. Alice receives a score of $-1$ if and only if she reports different signals on the two questions and Bob disagrees on both. Under truth-telling, this happens with probability $\Pr(S^{CA}(\tau) = -1) = 2\Pr(X_a = 0, X_b = 1)\Pr(X_a = 1, X_b = 0)$. However, if Alice always reports the same signal (e.g., $\mu(\sigma) = 0$ for any $\sigma$), she can avoid this outcome entirely. Therefore, $\mu$ is not stochastically dominated by $\tau$.

This warns that **repeatedly using the same question as the bonus or the penalty question will sabotage the SD-truthfulness of CA.**

## 5 A Rounding Reduction For Stochastically Dominant-Truthfulness

We introduce a straightforward rounding reduction that maps any truthful mechanism to an SD-truthful one. However, this approach often comes at a huge cost of sensitivity. To address this, we propose a more refined rounding method that partially restores the original mechanism's sensitivity.

### 5.1 Direct Rounding

Consider a peer prediction mechanism $\mathcal{M}$ that assigns Alice a score $S^{\mathcal{M}}$ with bounded support $S_{\sup}^{\mathcal{M}} < \infty$ and $S_{\inf}^{\mathcal{M}} > -\infty$. In *direct rounding*, we normalize the score to a probability: $\lambda^{\mathcal{M}} = \frac{S^{\mathcal{M}} - S_{\inf}^{\mathcal{M}}}{S_{\sup}^{\mathcal{M}} - S_{\inf}^{\mathcal{M}}} \in [0, 1]$. Alice's final score $\tilde{S}^{\mathcal{M}}$ is then determined by a binary lottery: receiving 1 with probability $\lambda^{\mathcal{M}}$ and 0 otherwise.

**Proposition 5.1.** *The direct rounding reduction of a truthful mechanism with bounded scores is SD-truthful.*

Intuitively, because direct rounding always outputs a Bernoulli score, SD-truthfulness simply implies that truth-telling should maximize the expected success probability. But the expected success probability is just the success probability. By our design of $\lambda^{\mathcal{M}}$, the success probability is a linear function of the final score $S^{\mathcal{M}}$, which proves equivalence between SD-truthfulness and truthfulness.

However, scoring agents via a single binary score is clearly not practical. The following proposition suggests that direct rounding usually destroys the sensitivity of the original mechanism. Let $\delta^{\mathcal{M}}(e)$

and $\tilde{\delta}^{\mathcal{M}}(e)$ be the sensitivity of $\mathcal{M}$ and its direct rounding reduction. Additionally, let $m^{\mathcal{M}}(e)$ and $\mathrm{std}^{\mathcal{M}}(e)$ be the expected score and the standard deviation.

**Proposition 5.2.** *The sensitivity ratio between a mechanism and its direct rounding is:*

$$\frac{\delta^{\mathcal{M}}(e)}{\tilde{\delta}^{\mathcal{M}}(e)} = \frac{\sqrt{\left(S_{\sup}^{\mathcal{M}} - m^{\mathcal{M}}(e)\right)\left(m^{\mathcal{M}}(e) - S_{\inf}^{\mathcal{M}}\right)}}{std^{\mathcal{M}}(e)}.$$

This ratio is typically large. For example, the final score of OA is an averaged Binomial $\mathrm{Bin}(n, m^{OA}(e))$, whose standard deviation is $\sqrt{m^{OA}(e)(1 - m^{OA}(e))/n}$. Thus, the sensitivity ratio between OA and its direct rounding is $\sqrt{n}$. This illustrates a key drawback of direct rounding: it greatly reduces the sensitivity by eliminating the benefit of averaging over multiple questions.

## 5.2 Partition Rounding

To reduce variance of and improve sensitivity, we aim to better leverage the information from all $n$ questions. In particular, if the final score of $S^{\mathcal{M}}$ is the average of $K$ independent individual scores $S_i^{\mathcal{M}}$, we can apply direct rounding to each $S_i^{\mathcal{M}}$ and take the average. This motivates the idea of *partition rounding*. We first partition the $n$ questions into $K$ disjoint subsets, using each to compute an individual score. Each score is then directly rounded as in Section 5.1, and the final score $\hat{S}^{\mathcal{M}}$ is the average of the $K$ rounded scores. The following lemma proves the feasibility of this idea.

**Lemma 5.3.** *Let $S = \frac{1}{K}\sum_{i\in[K]} S_i$, where the individual scores $S_i$ are i.i.d.. Then, $S(\tau)$ first-order stochastically dominates $S(\theta)$ if and only if each individual score $S_i(\tau)$ first-order stochastically dominates $S_i(\theta)$.*

**Proposition 5.4.** *The partition rounding reduction of a truthful mechanism with bounded scores is SD-truthful.*

We defer the proof of Lemma 5.3, which follows from a straightforward coupling argument, to Appendix F.3. Proposition 5.4 then follows directly from this lemma and the SD-truthfulness of direct rounding.

We now show that the partition rounding reduction can partially recover the sensitivity of the original mechanism. The key observation is that the final score under partition rounding is the average of a binomial variable, i.e. $\hat{S}^{\mathcal{M}}(e) = \frac{1}{K}\mathrm{Bin}(K, \lambda^{\mathcal{M}}(e))$, where $\lambda^{\mathcal{M}}(e) = \frac{m^{\mathcal{M}} - S_{\inf}^{\mathcal{M}}}{S_{\sup}^{\mathcal{M}} - S_{\inf}^{\mathcal{M}}}$ with $m^{\mathcal{M}}$ being the expectation of the individual score. Putting this into Definition 3.3 gives us the sensitivity of the partition rounding reduction

$$\hat{\delta}^{\mathcal{M}}(e) = \frac{\nabla m^{\mathcal{M}}(e) \cdot \sqrt{K}}{\sqrt{\left(m^{\mathcal{M}}(e) - S_{\inf}^{\mathcal{M}}\right)\left(S_{\sup}^{\mathcal{M}} - m^{\mathcal{M}}(e)\right)}}. \tag{1}$$

The sensitivity of the partition rounding reduction is exactly $\sqrt{K}$ times that of the direct rounding reduction. This means that by using partition rounding, we can recover a substantial portion of the original mechanism's sensitivity.

In Appendix B, we present the sensitivities of the partition-rounded versions of the three mechanisms discussed. We summarize the limitations of these mechanisms below which motivates the design of a new SD-truthful mechanism introduced in the next section.

- OA is already SD-truthful without rounding and thus has a high sensitivity. However, OA is SD-truthful only for self-dominating signals.

- PTS requires rounding to ensure SD-truthfulness. Therefore, when the prior of signals is biased, $S_{\sup}^{PTS}$ becomes large, which in turn leads to a low sensitivity.

- CA uses two questions to compute an individual score, so $K = n/2$. Furthermore, only bonus questions contribute to sensitivity, while penalty questions are used to enforce truthfulness. This dilutes the sensitivity by half again. Therefore, only $1/4$ of the questions count for the sensitivity of the partition-rounded CA. Hence, even without rounding, the sensitivity of the partition-rounded CA is only half of that of the original implementation.

# 6 The Enforced Agreement Mechanism

This section aims to present a novel mechanism with high sensitivity, which does not require the above rounding reduction. To motivate our idea, we begin by summarizing the key lessons learned from the previous discussions.

First, Example 4.1 shows a particular type of strategy that prevents previous mechanisms from being SD-truthful: reducing/changing the variance of scores, e.g. by always reporting the same signal. This inspires our *enforced agreement (EA)* mechanism, where the idea is to control a randomness level within agents' reports so that strategic manipulations cannot arbitrarily alter the variance of the scores.

Second, our analysis of sensitivity in Section 5.2 suggests that a high-sensitivity mechanism requires a larger number of partitions $K$ and a smaller score range $S_{\sup} - S_{\inf}$. This insight guides us to use just one question to compute an individual score (so $K = n$) and avoid the need for rounding. We show that EA meets these criteria.

We first prove the SD-truthfulness of EA in the binary-signal setting, and then establish a negative result explaining why it fails to ensure SD-truthfulness when signals are non-binary. Nonetheless, we show that EA remains truthful in non-binary settings when the signal structure is known or can be accurately learned.

## 6.1 EA in the Binary-signal Setting

We introduce EA in the binary-signal setting and defer the discussion for general settings to the appendix. The idea of EA is to control the marginal distribution of Alice's reports. In the binary setting, the mechanism commits to a target empirical distribution $\Phi = (n_0, n_1)$ where $n_0 + n_1 = n$. The mechanism expects Alice to report signal $i$ on exactly $n_i$ out of $n$ questions. If Alice's actual distribution $\Phi_{X_a} = (m_0, m_1)$ and suppose W.L.O.G. that $m_0 > n_0$, the mechanism randomly selects $m_0 - n_0$ of her 0-reports and flips them to 1. After enforcing $\Phi$, Alice is scored using the output agreement mechanism (OA).

Compared with OA, the enforcement process in EA prevents Alice from benefiting by over-reporting the majority signal. However, this enforcement introduces dependencies across questions, so the binary agreement scores are no longer i.i.d., and Lemma 5.3 does not apply. Our main result shows that EA is SD-truthful when signals are self-prediction (Definition 4.3).

**Theorem 6.1.** *If $|\Sigma| = 2$ and signals are self-predicting, the enforced agreement mechanism is SD-truthful for any $\Phi$.*

We defer the detailed proof to the appendix and present the main ideas below. Note that the final score of the enforced agreement mechanism is the average of $n$ Bernoulli variables, each with a potentially different success probability. In the binary-signal setting, there are four possible success probabilities in total—$p_{ij} = \Pr(X_b = j \mid X_a = i)$ for $i, j \in \{0, 1\}$—representing the probability of Bob observing (and reporting) $j$ on a question conditioned on Alice observing $i$. A Bernoulli variable with success probability $p_{ij}$ corresponds to a question on which Alice observes $i$ but is flipped to $j$ either by the mechanism or by Alice herself. Suppose W.L.O.G. that $m_0 > n_0$, meaning that the mechanism will randomly flip $m_0 - n_0$ reports of Alice from 0 to 1 under truth-telling. In this case, the final score of Alice is the average of three binomials:

$$n \cdot S^{EA}(\tau) \sim \mathrm{Bin}(n_0, p_{00}) + \mathrm{Bin}(m_1, p_{11}) + \mathrm{Bin}(m_0 - n_0, p_{01}). \tag{2}$$

We show that for any untruthful strategy, the resulting final score can be written as:

$$n \cdot S^{EA}(\theta) \sim \mathrm{Bin}(n_0 - k, p_{00}) + \mathrm{Bin}(m_1 - k, p_{11}) + \mathrm{Bin}(m_0 - n_0 + k, p_{01}) + \mathrm{Bin}(k, p_{10}),$$

where $0 \le k \le \min(n_0, m_1)$. Under the self-prediction condition, $p_{00} > p_{10}$ and $p_{11} > p_{01}$. Therefore, any untruthful strategy can only reallocate $k$ samples from two binomial distributions with higher success probabilities to those with smaller success probabilities. Via a simple coupling argument, we can see that $S(\theta)$ is first-ordered stochastically dominated by $S(\tau)$.

Additional results are provided in Appendix C. First, we show that EA is not generally SD-truthful in the non-binary setting. Strategic signal permutations can produce score distributions that are not stochastically dominated by truth-telling. However, EA remains truthful when the information structure is known or well estimated. Moreover, we derive a closed-form expression for the sensitivity-maximizing $\Phi$, enabling computation of the optimal enforcement from the information structure.

# 7 Empirical Evaluations of SD-Truthful Mechanisms

In this section, we empirically compare the sensitivity of the proposed mechanisms across various settings. The results support our theoretical findings: EA consistently exhibits the highest sensitivity among SD-truthful mechanisms in the binary-signal setting (except when the signal prior is nearly uniform).

## 7.1 Datasets and Experiment Setup

We use two real-world datasets to estimate the information structure between two agents. The first dataset contains binary labels classifying whether a compound is appropriate or inappropriate to be synthesized [Baba et al., 2018]. The second dataset collects the annotations of the sentiment of 300 tweets, where the size of the signal space is 4 [Venanzi et al., 2015]. We denote these two information structures as $J_1$ and $J_2$ respectively, and defer the details to Appendix E.1.

In the main body of the paper, we focus on SD-truthful mechanisms, which include:

- **OA** (Section 4.1), which is SD-truthful only when signals are self-dominating.
- **PTS-partition-round**—the partition rounding reduction of PTS (Section 4.2), where we re-weight every individual score using signal prior and take the average.
- **CA-partition-round**—the partition rounding reduction of CA (Section 4.3), where we create $K = n/2$ partitions of questions and score agents the average of the individual CA score for each partition after rounding.
- **MA-partition-round**—the partition rounding reduction of the matching agreement mechanism (Appendix D).
- **EA-prior**—EA with enforcement $\Phi = n \cdot \Pr(X_a)$.
- **EA-uniform**—EA with $\Phi_i = n \cdot 1/|\Sigma|$, $\forall i \in \Sigma$.
- **EA-optimal**—EA with the sensitivity-maximizing $\Phi$ computed according to Appendix C.3.

For information structures with binary-signal settings, sensitivities can be computed analytically as shown in Appendix B. For general cases, we estimate sensitivity using a Monte Carlo approach. In particular, we simulate reports to each of the $n$ questions with both agents exerting full effort $e = 1$ and run $\mathcal{M}$ to compute the scores. Repeating this process for $T = 20{,}000$ times yields $T$ i.i.d. samples of $S^{\mathcal{M}}(e)$. We then compute the score of Alice when she deviates to a lower effort level at $e - \Delta e = 0.8$, and obtain $T$ samples of $S^{\mathcal{M}}(e - \Delta e)$. The sensitivity of at effort $e$ is then estimated as $\frac{\mathbb{E}[S^{\mathcal{M}}(e)] - \mathbb{E}[S^{\mathcal{M}}(e - \Delta e)]}{\Delta e} \cdot \frac{1}{\text{std}(S^{\mathcal{M}}(e))}$.

## 7.2 The Sensitivity of SD-Truthful Mechanisms

We now compare the sensitivity of the discussed SD-truthful mechanisms. In Figure 1a, we show the sensitivity when the prior of the binary signal is varied, i.e. $\Pr[X_a = 0] \in [0.1, 0.9]$, and the number of questions $n = 100$.

Figure 1b and 1c present sensitivity as the number of questions $n \in \{10, 20, \ldots, 100\}$ increases, under information structures $J_1$ and $J_2$, respectively. In $J_2$, where $|\Sigma| > 2$, EA is no longer SD-truthful; however, we still include its sensitivity for comparison.

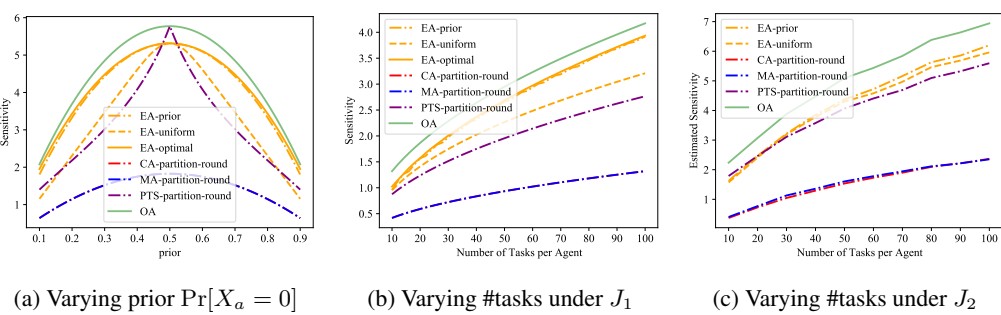

(a) Varying prior $\Pr[X_a = 0]$     (b) Varying #tasks under $J_1$     (c) Varying #tasks under $J_2$

Figure 1: A Comparison of the Sensitivity of Different Mechanisms.

We make the following observations. First, although the partition reductions of CA and MA are SD-truthful for general settings, their sensitivities are dominated by other SD-truthful mechanisms. This is due to their requirements of the penalty questions, which do not contribute to the sensitivity, and the fact that they require two questions to compute an individual score.

Second, PTS performs well when the signal prior is nearly uniform, i.e. $\Pr(X_a = 0) \approx 0.5$. Intuitively, this is because the rounding factor that enlarges the gap $S_{\sup}^{PTS} - S_{\inf}^{PTS}$ shrinks when the prior is close to uniform. However, even a slight bias, i.e. $|\Pr(X_a = 0) - 0.5| > 0.03$ causes PTS-partition-round to be outperformed by EA.

Third, we observe that enforcing the signal prior in EA closely approximates the optimal enforcement. Remarkably, even EA-uniform, a detail-free version that doesn't require knowledge of the prior, consistently outperforms PTS-partition-round, which does rely on prior knowledge for SD-truthfulness.

Additionally, we note that the partition-rounding reduction of MA coincides with that of CA when signals are binary (see Appendix D). Consequently, the blue and red overlap in Figure 1a and 1b.

Due to space constraints, we defer the comparison between partition-rounding reductions and their original implementations to Appendix E.2. These results further show that high-sensitivity mechanisms perform strongly in practical, incentive-aligned environments, underscoring the importance of EA.

## 8    Conclusion and Future Work

This paper initiates the discussions of stochastically dominant-truthful (SD-truthful) peer prediction mechanisms, which ensure that in equilibrium, the score distribution under truth-telling first-order stochastically dominates any other strategy. Unlike traditional peer prediction mechanisms, which rely on linear utility assumptions, SD-truthful mechanisms can preserve truthfulness for any increasing utility function. We show that existing mechanisms fail to naturally achieve SD-truthfulness and propose a rounding method to enforce it. However, this process often reduces the sensitivity of the original mechanism, making it less efficient in practice. Additionally, we introduce the Enforcement Agreement (EA) mechanism, which is empirically shown to have the highest sensitivity among all SD-truthful mechanisms in the binary-signal setting—except when the signal prior is nearly uniform.

Future work is needed to investigate more sensitive SD-truthful mechanisms beyond the binary-signal setting. A promising next step is to explore approximate SD-truthfulness, which allows truth-telling to lose a small utility for some utility functions. This relaxation could lead to mechanisms with better sensitivity. Furthermore, SD-truthfulness alone does not ensure that truth-telling maximizes expected rewards in a tournament (rank-order rewards). Because agents' scores are correlated, a manipulation strategy with a dominated score distribution may still raise an agent's chance of ranking above others by altering these correlations. An important direction for future work is therefore the design of peer prediction mechanisms that provide incentive guarantees under tournaments.

## Broader Impact

On the positive side, our method enhances the reliability and efficiency of crowdsourced data collection, which supports more trustworthy machine learning systems and reduces reliance on expert annotation. However, mechanisms that incentivize effort may unintentionally disadvantage contributors with limited time, resources, or familiarity with the task, raising fairness concerns. We aim to mitigate these risks by designing robust, broadly applicable mechanisms and promoting transparency in their deployment.

## Acknowledgment

This work was partly carried out at the DIMACS Center at Rutgers University and supported by NSF #2313137.

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

## A  Continued Related Work: Mutual Information Mechanisms

We review several classic peer prediction mechanisms that evaluate agents by estimating mutual information, and illustrate why these mechanisms fail to achieve SD-truthfulness. First, proposed by Kong and Schoenebeck [2019], we can estimate the empirical point-wise joint distribution between $\hat{X}_{a,\cdot}$ and $\hat{X}_{b,\cdot}$, using pairs of reports from two agents, $(\hat{X}_{a,j}, \hat{X}_{b,j})_{j \in [n]}$. Note that because questions are i.i.d., we can use reports from all questions to estimate the same distribution. Then, plugging the learned empirical joint distribution into an $f$-divergence operator returns a (biased) estimate of the $f$ point-wise mutual information. The following is the score for the $f$-mutual information mechanism [Kong and Schoenebeck, 2019]:

$$S^{fMI} = D_f \left( \Pr \left( \hat{X}_{a,\cdot}, \hat{X}_{b,\cdot} \right) \| \Pr \left( \hat{X}_{a,\cdot} \right) \Pr \left( \hat{X}_{b,\cdot} \right) \right).$$

However, because the estimation is biased, the $f$-mutual information mechanism is only asymptotically truthful, meaning that it requires an infinite number of i.i.d. questions to satisfy Definition 3.1. When the number of tasks is small, there is a non-zero probability that truth-telling is not the best response. For example, if Alice observes "yes" on all her questions, which happens with a positive probability, then she knows that truth-telling will always result in a score of 0 regardless of Bob's responses. This is because the estimated point-wise joint distribution is the same as the product of marginal distributions. However, by misreporting "no" on some questions, she has a non-zero probability of receiving a positive score. In comparison, such deviations are discouraged under the enforced agreement (EA) mechanism because the mechanism will randomly flip the responses on behalf of the agent (see Section 6).

Several follow-up works, such as the Determinant Mutual Information (DMI) mechanism by Kong [2020] and the pairing mechanism by Schoenebeck and Yu [2020], propose different ideas to estimate the mutual information. However, they are still not SD-truthful. Consider the strategy from Example 4.1, where Alice always reports "yes." Under both DMI and the pairing mechanism, this strategy always yields a score of zero. In contrast, similar to the CA mechanism, there is a positive probability that truth-telling may result in a negative score. Thus, this constant-reporting strategy is not dominated by truth-telling under these mutual information-based mechanisms whose score can take negative values.

## B  Sensitivity of Prior Mechanisms Under Partition Rounding

We now map the exemplary mechanisms discussed in Section 4 into the partition rounding framework. In particular, we specify the key parameters $K$, $S_{\inf}$, $S_{\sup}$, and $m_i(e)$ for each mechanism. Consider an information structure with joint distribution $J$, and the marginal distribution of Alice and Bob being $M^a$ and $M^b$, respectively. Furthermore, let $M_{\sigma,\sigma'} = M_\sigma^a M_{\sigma'}^b$ denote the product of marginal distributions.

**Output Agreement**  An individual score of OA, $S_i^{OA}$, is 1 if both agents agree on a question $i$ and 0 otherwise. Therefore, OA is the partition rounding reduction of itself with $K = n$, $S_{\inf} = 0$, and $S_{\sup} = 1$. By definition, when Alice exerts effort, agents' signals on the same question is sampled from the joint distribution $J$; while if Alice does not exert effort, she will sample a signal from the prior, meaning that agents' signals on the same question is sampled from $M$. The expectation of an individual score at the effort level $e$ is thus

$$m_i^{OA}(e) = e \cdot \Pr(X_{a,i} = X_{b,i}) + (1 - e) \cdot \Pr(\mu(X_{a,i}) = \mu(X_{b,i})) = e \cdot \mathrm{tr}(J) + (1 - e) \cdot \mathrm{tr}(M).$$

This means that the derivative of the expected score w.r.t. effort $e$ is

$$\nabla m_i^{OA}(e) = \mathrm{tr}(J - M) = \mathrm{tr}(\Delta),$$

where $\Delta$ is the difference between the joint distribution and the product of marginal distributions, and $\mathrm{tr}(\cdot)$ represents the trace of a matrix, i.e. the sum of the diagonal entries.

**Peer Truth Serum**  Similar to OA, each partition of PTS is composed of a single question so that $K = n$, $S_{\inf} = 0$, and $S_{\sup} = \frac{1}{M_{\sigma_{\min}}^a}$ where $\sigma_{\min}$ is the signal occurs with the smallest probability in prior. The expected individual score is

$$m_i^{PTS}(e) = \sum_{\sigma \in \Sigma} e \cdot \frac{J_{\sigma,\sigma}}{M_\sigma^a} + (1 - e) \cdot M_\sigma^b = 1 + e \cdot \sum_{\sigma \in \Sigma} \frac{\Delta_{\sigma,\sigma}}{M_\sigma^a}; \qquad \nabla m_i^{PTS}(e) = \sum_{\sigma \in \Sigma} \frac{\Delta_{\sigma,\sigma}}{M_\sigma^a}.$$

**Correlated Agreement**  The CA mechanism requires two questions to compute an individual score, meaning that $K = n/2$. By Proposition F.1 and Lemma 5.3, when the signal space is binary, scoring Alice using $S^{CA} = \sum_{i \in [K]} S_i^{CA}$ without rounding is SD-truthful. However, when $|\Sigma| > 2$, we have to use the partition rounding reduction of CA to obtain SD-truthful where $S_{\inf} = -1$ and $S_{\sup} = 1$. On the bonus question, the probability of receiving a score of 1 is given by $\sum_{\sigma,\sigma'} (e \cdot J_{\sigma,\sigma'} + (1 - e) \cdot M_{\sigma,\sigma'}) T_\Delta(\sigma, \sigma')$. On the penalty question, the probability of penalizing a score of 1 is given by $\sum_{\sigma,\sigma'} (e \cdot M_{\sigma,\sigma'} + (1 - e) \cdot M_{\sigma,\sigma'}) T_\Delta(\sigma, \sigma')$. Combining these,

$$m_i^{CA}(e) = e \cdot \sum_{\sigma,\sigma' \in \Sigma} \Delta_{\sigma,\sigma'} \, T_\Delta(\sigma, \sigma') = e \cdot \sum_{\sigma,\sigma' \in \Sigma} (\Delta_{\sigma,\sigma'})^+; \qquad \nabla m_i^{CA}(e) = \sum_{\sigma,\sigma' \in \Sigma} (\Delta_{\sigma,\sigma'})^+,$$

where $(x)^+ = x$ if $x > 0$ and 0 otherwise. Putting these parameters into Equation (1) returns the sensitivity of the partition rounding reduction for CA. However, note that by Proposition F.1, we do not need to round the individual CA scores when $|\Sigma| = 2$. This motivates a variant of an SD-truthful implementation of CA that may have a slightly larger sensitivity in the binary-signal setting. We will introduce and test this implementation in Section 7.

We summarize the limitations of the above three SD-truthful mechanisms.

- OA does not require the partition rounding reduction to achieve SD-truthfulness and thus is likely to have a high sensitivity. However, OA is SD-truthful only when signals are self-dominating.

- PTS requires rounding to ensure SD-truthfulness. The normalizing factor $\frac{1}{M_{\sigma_{\min}}^a}$ implies that PTS will be less sensitive when the prior distribution of signals is biased.

- For the CA mechanism, only bonus questions contribute to the sensitivity of the mechanism, while penalty questions are used to enforce truthfulness. However, CA requires two questions to compute a single individual score, which means $K = n/2$.[3] Therefore, even without rounding, e.g. when $|\Sigma| = 2$, the number of effective questions is only one half of the available questions. Moreover, the rounding process divides each score by a factor of 2, which further harms the sensitivity of the mechanism.

## C  The Enforced Agreement Mechanism (Continued)

### C.1  Arbitrary Signal Space

Unfortunately, we show that the above idea does not generalize beyond the binary-signal setting. For a general setting with $|\Sigma| = c$, the enforced agreement mechanism can be characterized by an

---

[3]We emphasize that increasing $K$ by rerunning partitions is not SD-truthful (Example 4.1).

*enforced marginal distribution* $\Phi = (n_0, \ldots, n_{c-1})$ and an *enforcement rule* that ensures an agent's reports to follow $\Phi$.

Let Alice's report vector $\hat{X}_a$ have an empirical marginal distribution $\Phi_{\hat{X}_a} = (\hat{m}_0, \ldots, \hat{m}_{c-1})$. The enforcement rule can be characterized as a $c \times c$ matrix $\rho\left(\hat{X}_a\right)$ (or simply $\rho$ if there is no confusion), such that $\rho_{i,j}$ denotes the number of randomly selected questions on which Alice reports $i$ and is flipped to $j$ by the mechanism. By construction, $\rho_{i,j} \in \mathbb{Z}_{\geq 0}$ for any $i, j \in \Sigma$. Furthermore, $\sum_i \rho_{i,j} = n_j$ for any $j \in \Sigma$ and $\sum_j \rho_{i,j} = \hat{m}_i$ for any $i \in \Sigma$.

Suppose Alice's signal vector $X_a$ has a marginal distribution $\Phi_a = (m_0, \ldots, m_{c-1})$. Under the enforcement rule, while reasoning the distribution the of EA score, every (randomized) strategy can be characterized by (a mixture of) *manipulation matrix*. The entry $\vartheta_{i,j}$ of a manipulation matrix denotes the number of randomly selected questions where Alice observes $i$ but is flipped to $j$ either by the mechanism or by herself. Similarly, $\vartheta_{i,j} \in \mathbb{Z}_{\geq 0}$ for any $i, j \in \Sigma$, $\sum_i \vartheta_{i,j} = n_j$ for any $j \in \Sigma$, and $\sum_j \vartheta_{i,j} = m_i$ for any $i \in \Sigma$. In particular, let $\vartheta^\tau(\rho)$ denote the manipulation matrix corresponding to truth-telling under the enforcement rule $\rho$.

Given a manipulation matrix $\vartheta$, the final score of Alice is thus the average of $c^2$ binomials:

$$S^{EA}(\theta) \sim \frac{1}{n} \sum_{i,j \in \Sigma} \mathrm{Bin}(\vartheta_{i,j}, p_{ij}),$$

where $p_{ij} = \Pr(X_b = j \mid X_a = i)$ is the conditioned probability of agents' signals.

To obtain SD-truthfulness, we have to construct an enforcement rule $\rho$ such that for any $p$ (under some assumptions), any $X_a$, and any feasible $\vartheta$, the random variable $S^{EA}(\theta)$ is first-order stochastically dominated by $S^{EA}(\tau)$. This is particularly challenging when $c$ is large because $\vartheta$ has $c^2 - 2c + 1$ degrees of freedom, yet we must identify a $\rho$ such that $\vartheta^\tau(\rho)$ dominates any other $\vartheta$.

**Three signals**  We show that even for the three-signal setting, EA is SD-truthful only under a very special type of information structure.

**Definition C.1.** Agents' signals are uniformly self-predicting if $\Pr(X_b = i \mid X_a = i) > \Pr(X_b = i \mid X_a = j) = \Pr(X_b = i \mid X_a = k)$ for any $j \neq i$ and $k \neq i$.

Consider the following enforcement rule:

- *Alice over-reports one signal.* If $\hat{m}_i \geq n_i$ while $\hat{m}_j \leq n_j$ for any $j \neq i$, the enforcement rule will randomly select $n_j - \hat{m}_j$ questions on which Alice reports $i$ and flip them to $j$ for $j \in \Sigma$ and $j \neq i$.

- *Alice over-reports two signals.* If $\hat{m}_j < n_j$ while $\hat{m}_i \geq n_i$ for any $i \neq j$, the enforcement rule will randomly select $\hat{m}_i - n_i$ questions on which Alice reports $i$ and flip them to $j$ for $i \in \Sigma$ and $i \neq j$.

This enforcement rule flips the minimum number of reports to enforce $\Phi$ and thus is named the *minimal enforcement rule*, denoted as $\rho^*$.

**Proposition C.2.** *If $|\Sigma| = 3$ and signals are self-predicting, the enforcement agreement mechanism is SD-truthful for any $\Phi$ if and only if it applies the minimal enforcement rule and signals are uniformly self-predicting.*

We defer the proof to Appendix F.5. Our results imply that achieving SD-truthfulness using the EA mechanism is infeasible for arbitrary information structures beyond the binary-signal setting. Investigating alternative approaches, such as mechanism designs that leverage additional structure in the information space or stronger assumptions on agent strategies, remains an interesting direction for future work.

## C.2  Truthfulness of EA

Although the EA mechanism is not SD-truthful when $|\Sigma| > 2$, we show that it is still possible to obtain truthfulness as long as the information structure is known.

Given a report vector of Alice $\hat{X}_a$ with an empirical distribution $\Phi_{\hat{X}_a} = (\hat{m}_0, \dots, \hat{m}_{c-1})$, EA will apply the *IP enforcement rule* by solving the following integer programming.

$$\max_{\vartheta} \quad \sum_{i,j \in \Sigma} \vartheta_{i,j} \, p_{ij}$$

$$\text{s.t.} \quad \sum_{i \in \Sigma} \vartheta_{i,j} = n_j, \quad \forall j \in \Sigma,$$

$$\sum_{j \in \Sigma} \vartheta_{i,j} = \hat{m}_i, \quad \forall i \in \Sigma,$$

$$\vartheta_{i,j} \in \mathbb{Z}_{\geq 0}, \quad \forall i, j \in \Sigma.$$

Suppose the optimal solution is $\vartheta^*(\hat{X}_a)$. The mechanism will then randomly select $\vartheta^*(\hat{X}_a)_{i,j}$ questions on which $\hat{X}_a = i$ and flip them to $j$, for any $i, j \in \Sigma$. After the enforcement step, Alice will be scored based on the output agreement mechanism.

**Proposition C.3.** *For any enforcement $\Phi$ and information structure, if $\Pr(X_b \mid X_a)$ is known, the enforcement agreement mechanism with the IP enforcement rule is truthful.*

The proof is straightforward. Note that to prove truthfulness, it is sufficient to show that truth-telling maximizes the expected score which is $\sum_{i,j \in \Sigma} \vartheta_{i,j}^{\tau} \, p_{ij}$. Because the mechanism will manipulate agents' reports in an optimal way by solving the above integer programming, no strategy can achieve a higher expected score than truth-telling.

Note that when $|\Sigma| = 2$ and signals are self-predicting, the optimal solution to the above IP does not depend on the information structure $p_{ij}$. In particular, the IP enforcement rule always flips the over-reported signal to enforce $n_0$ zeros and $n - n_0$ ones and does not flip any of the under-reported signal. This echos Theorem 6.1 which suggests the EA mechanism is SD-truthful in the binary-signal setting without the knowledge of $p$.

*Remark* C.4 (Connection to Peer Truth Serum Faltings et al. [2017]). As illustrated in Section 4.2, PTS can be viewed as an output agreement mechanism (Section 4.1) where the score for agreement is weighted inversely by each signal's prior. This re-weighting shifts the probability of agreement from the joint distribution $\Pr(X_a, X_b)$, which demands self-dominating signals to ensure truthful reporting, to the conditional distribution $\Pr(X_b \mid X_a)$, which requires self-predicting signals.

In contrast, EA avoids using the prior $\Pr(X_a)$ by controlling the marginal distribution of Alice's reports. This approach similarly transforms the probability of agreement from the joint to the conditional distribution. However, the enforcement process creates additional incentive issues when the signal space is large which requires complete knowledge of the underlying information structure. Consequently, if $|\Sigma| = 2$ and signals are self-predicting, EA can obtain a stronger truthful guarantee (SD-truthfulness) without any prior knowledge; but for $|\Sigma| > 2$, it requires more extensive structural information to guarantee truthfulness.

### C.3 The Optimal Enforcement

We have established that when signals are binary and self-predicting, EA with any enforcement $\Phi$ is SD-truthful. In this subsection, we examine how the choice of enforcement, $\Phi = (n_0, n - n_0)$, influences the sensitivity of EA, and in particular, what is the sensitivity-maximizing enforcement.

Fixing an information structure, the expected score given by EA, $S^{EA}(n_0, e)$, can be viewed as a function of the number of questions where the answers are enforced to zero, $n_0$, and Alice's effort level, $e$. We first present an intermediate result suggesting that the standard deviation of $S^{EA}$ is independent of $n_0$. This implies that the enforcement affects the sensitivity of EA only through $\nabla \mathbb{E}[S^{EA}(e)]$.

**Lemma C.5.** *For a fixed information structure, $\mathrm{std}\left(S^{EA}(n_0, e)\right)$ is independent of $n_0$, i.e. $\mathrm{std}\left(S^{EA}(n_0, e)\right) = \mathrm{std}\left(S^{EA}(n_0', e)\right)$ for any $n_0, n_0' \in \{0, 1, \dots, n\}$.*

We next analyze how the derivative of the expected score depends on the choice of enforcement. We find that the sensitivity-maximizing enforcement $n_0$ is determined by a quantile of a binomial distribution with success probability $\Pr(X_a = 0)$, where the quantile depends on the information structure.

**Proposition C.6.** *If $|\Sigma| = 2$ and signals are self-predicting, the sensitivity-maximizing enforcement of the EA mechanism $\Phi = (n_0, n - n_0)$ satisfies that*

$$n_0 = \max \left\{ k \in \{0, 1, \ldots, n\} : F(k) < \frac{\eta'_{00} - \eta'_{01}}{\eta'_{00} - \eta'_{01} + \eta'_{11} - \eta'_{10}} \right\},$$

*where $F(k) = \sum_{i=0}^{k} \binom{n}{i} \Pr(X_a = 0)^i (1 - \Pr(X_a = 0))^{n-i}$ is the c.d.f. of a binomial random variable with success probability $\Pr(X_a = 0)$, and $\eta'_{ij} = \Pr(X_b = j \mid X_a = i) - \Pr(X_b = j)$.*

We defer the proofs of Lemma C.5 and Proposition C.6 to Appendix F.6.

## D   The Matching Agreement Mechanism

The matching agreement mechanism is a follow-up of the correlated agreement mechanism aiming to address more complex cheating strategies Zhang and Schoenebeck [2023b]. Similar to CA, which determines "agreement" using the delta matrix (see Section 4.3), MA determines "agreement" using a different method depending on the information structure.

**Definition D.1.** Let $\Gamma$ be a $|\Sigma| \times |\Sigma| \times |\Sigma|$ tensor where

$$\Gamma_{\sigma_1, \sigma_2, \sigma_3} = \Pr(X_a = \sigma_1 \mid X_b = \sigma_2) - \Pr(X_a = \sigma_1 \mid X_b = \sigma_3).$$

The agreement function is thus $T_\Gamma = \mathrm{Sign}(\Gamma)$, i.e. $T_\Gamma(\sigma_1, \sigma_2, \sigma_3) = 1$ if $\Gamma_{\sigma_1, \sigma_2, \sigma_3} > 0$, and 0 otherwise.

Next, the mechanism randomly samples a bonus question $j$ and a penalty question $k$ uniformly at random. An individual score computed using these two questions is thus $S_i^{MA} = T_\Gamma(\hat{X}_{a,j}, \hat{X}_{b,j}, \hat{X}_{b,k})$. In words, the matching agreement mechanism awards Alice a score of 1 if, when predicting her response on question $b$, the posterior conditioned on Bob's response to the same question provides a better prediction than the posterior conditioned on Bob's response on a different question $q$. Similar to CA, each bonus question can be paired with $n - 1$ penalty questions. Therefore, the final score of MA is the average of $n \cdot (n - 1)$ individual scores.

**The Sensitivity of MA Under the Partition Rounding Reduction**   We further investigate the sensitivity of the MA mechanism under the partition reduction. The MA mechanism requires two questions to compute an individual score, implying that $K = n/2$. Clearly, $S_{\inf} = 0$ and $S_{\sup} = 1$ by definition. By Equation (1), to compute the sensitivity, we only need to compute the expected individual score $m_i(e)$ and its derivative. When Alice exerts full effort, the probability of obtaining a score of 1 can be computed as follows:

$$
\begin{aligned}
m_i^{MA}(1) &= \sum_{\sigma_1, \sigma_2, \sigma_3} \Pr(X_{a,j} = \sigma_1, X_{b,j} = \sigma_2, X_{b,k} = \sigma_3) \cdot T_\Gamma(\sigma_1, \sigma_2, \sigma_3) \\
&= \sum_{\sigma_1, \sigma_2, \sigma_3} J_{\sigma_1, \sigma_2} M_{\sigma_3}^b \cdot T_\Gamma(\sigma_1, \sigma_2, \sigma_3) \\
&= \frac{1}{2} \sum_{\sigma_1, \sigma_2, \sigma_3} J_{\sigma_1, \sigma_2} M_{\sigma_3}^b \cdot T_\Gamma(\sigma_1, \sigma_2, \sigma_3) + \frac{1}{2} \sum_{\sigma_1, \sigma_2, \sigma_3} J_{\sigma_1, \sigma_3} M_{\sigma_2}^b \cdot T_\Gamma(\sigma_1, \sigma_3, \sigma_2) \\
&= \frac{1}{2} + \frac{1}{2} \sum_{\sigma_1, \sigma_2, \sigma_3} \left( J_{\sigma_1, \sigma_2} M_{\sigma_3}^b - J_{\sigma_1, \sigma_3} M_{\sigma_2}^b \right) \cdot T_\Gamma(\sigma_1, \sigma_2, \sigma_3) \\
&\qquad\qquad\qquad \text{(Note that } T_\Gamma(\sigma_1, \sigma_2, \sigma_3) = 1 - T_\Gamma(\sigma_1, \sigma_3, \sigma_2).\text{)} \\
&= \frac{1}{2} + \frac{1}{2} \sum_{\sigma_1, \sigma_2, \sigma_3} M_{\sigma_2}^b M_{\sigma_3}^b \, \Gamma_{\sigma_1, \sigma_2, \sigma_3} \, T_\Gamma(\sigma_1, \sigma_2, \sigma_3) \\
&= \frac{1}{2} + \frac{1}{2} \sum_{\sigma_1, \sigma_2, \sigma_3} M_{\sigma_2}^b M_{\sigma_3}^b \, (\Gamma_{\sigma_1, \sigma_2, \sigma_3})^+
\end{aligned}
$$

where $(x)^+ = x$ if $x > 0$ and 0 otherwise.

When Alice exerts no effort,

$$m_i^{MA}(0) = \sum_{\sigma_1,\sigma_2,\sigma_3} \Pr(\hat{X}_{a,j} = \sigma_1, X_{b,j} = \sigma_2, X_{b,k} = \sigma_3) \cdot T_\Gamma(\sigma_1, \sigma_2, \sigma_3)$$

$$= \sum_{\sigma_1,\sigma_2,\sigma_3} M_{\sigma_1}^a M_{\sigma_2}^b M_{\sigma_3}^b \cdot T_\Gamma(\sigma_1, \sigma_2, \sigma_3)$$

$$= \frac{1}{2} \sum_{\sigma_1,\sigma_2,\sigma_3} M_{\sigma_1}^a M_{\sigma_2}^b M_{\sigma_3}^b \cdot T_\Gamma(\sigma_1, \sigma_2, \sigma_3) + \frac{1}{2} \sum_{\sigma_1,\sigma_2,\sigma_3} M_{\sigma_1}^a M_{\sigma_2}^b M_{\sigma_3}^b \cdot T_\Gamma(\sigma_1, \sigma_3, \sigma_2)$$

$$= \frac{1}{2} + \frac{1}{2} \sum_{\sigma_1,\sigma_2,\sigma_3} \left( M_{\sigma_1}^a M_{\sigma_2}^b M_{\sigma_3}^b - M_{\sigma_1}^a M_{\sigma_2}^b M_{\sigma_3}^b \right) \cdot T_\Gamma(\sigma_1, \sigma_2, \sigma_3)$$

(Note that $T_\Gamma(\sigma_1, \sigma_2, \sigma_3) = 1 - T_\Gamma(\sigma_1, \sigma_3, \sigma_2)$.)

$$= \frac{1}{2}$$

The expected score while exerting effort $e$ is a convex combination of $m_i^{MA}(1)$ and $m_i^{MA}(0)$, i.e. $m_i^{MA}(e) = \frac{1}{2} + \frac{e}{2} \sum_{\sigma_1,\sigma_2,\sigma_3} M_{\sigma_2}^b M_{\sigma_3}^b (\Gamma_{\sigma_1,\sigma_2,\sigma_3})^+$, and $\nabla m_i^{MA}(e) = \frac{1}{2} \sum_{\sigma_1,\sigma_2,\sigma_3} M_{\sigma_2}^b M_{\sigma_3}^b (\Gamma_{\sigma_1,\sigma_2,\sigma_3})^+$.

When signals are binary, we find that MA reduces to CA.

**Lemma D.2.** *When signals are binary, $\sum_{\sigma_1,\sigma_2,\sigma_3} M_{\sigma_2}^b M_{\sigma_3}^b (\Gamma_{\sigma_1,\sigma_2,\sigma_3})^+ = \sum_{\sigma_1,\sigma_2} (\Delta_{\sigma_1,\sigma_2})^+$.*

*Proof.* First note that by definition, $\Gamma_{\sigma_1,\sigma_2,\sigma_3} = 0$ when $\sigma_2 = \sigma_3$. Therefore, when signals are binary,

$$\sum_{\sigma_1,\sigma_2,\sigma_3} M_{\sigma_2}^b M_{\sigma_3}^b (\Gamma_{\sigma_1,\sigma_2,\sigma_3})^+ = \sum_{\sigma_1,\sigma_2} M_{\sigma_2}^b M_{1-\sigma_2}^b (\Gamma_{\sigma_1,\sigma_2,1-\sigma_2})^+$$

$$= \sum_{\sigma_1,\sigma_2,\sigma_3} M_{\sigma_2}^b M_{\sigma_3}^b (\Gamma_{\sigma_1,\sigma_2,\sigma_3})^+$$

$$= \sum_{\sigma_1,\sigma_2} (J_{\sigma_1,\sigma_2} M_{1-\sigma_2}^b - J_{\sigma_1,1-\sigma_2} M_{\sigma_2}^b)^+$$

$$= \sum_{\sigma_1,\sigma_2} (J_{\sigma_1,\sigma_2} (1 - M_{\sigma_2}^b) - ((M_{\sigma_1}^a - J_{\sigma_1,\sigma_2}) M_{\sigma_2}^b)^+$$

(Signals are binary.)

$$= \sum_{\sigma_1,\sigma_2} (J_{\sigma_1,\sigma_2} - M_{\sigma_1}^a M_{\sigma_2}^b)^+$$

$$= \sum_{\sigma_1,\sigma_2} (\Delta_{\sigma_1,\sigma_2})^+$$

$\square$

An immediate consequence of Lemma D.2 is that the expected score under MA is a linear transformation of the expected score under CA. This further implies that the partition rounding reductions of MA and CA has the same sensitivity under the binary-signal setting.

**Proposition D.3.** *When signals are binary, $\delta^{MA}(e) = \delta^{CA}(e)$ under the partition rounding reduction.*

*Proof.* By Lemma D.2, we know that

$$m_i^{MA}(e) = \frac{1}{2} + \frac{e}{2} \sum_{\sigma_1,\sigma_2,\sigma_3} M_{\sigma_2}^b M_{\sigma_3}^b (\Gamma_{\sigma_1,\sigma_2,\sigma_3})^+ = \frac{1}{2} + \frac{e}{2} \sum_{\sigma_1,\sigma_2} (\Delta_{\sigma_1,\sigma_2})^+ = \frac{1}{2} + \frac{1}{2} m_i^{CA}(e),$$

$$\nabla m_i^{MA}(e) = \frac{1}{2} \nabla m_i^{CA}(e).$$

By Equation (1),

$$
\begin{aligned}
\delta^{MA}(e) &= \frac{\nabla m_i^{MA}(e) \cdot \sqrt{n/2}}{\sqrt{\left(m_i^{MA}(e) - S_{\inf}^{MA}\right)\left(S_{\sup}^{MA} - m_i^{MA}(e)\right)}} \\
&= \frac{\frac{1}{2}\nabla m_i^{CA}(e) \cdot \sqrt{n/2}}{\sqrt{\left(\frac{1}{2}m_i^{MA}(e) + \frac{1}{2}\right)\left(\frac{1}{2} - \frac{1}{2}m_i^{MA}(e)\right)}} \\
&= \frac{\nabla m_i^{CA}(e) \cdot \sqrt{n/2}}{\sqrt{\left(m_i^{MA}(e) + 1\right)\left(1 - m_i^{CA}(e)\right)}} \\
&= \delta^{CA}(e).
\end{aligned}
$$

$\square$

# E  Experiments (Continue)

All codes for our experiments are provided in `https://github.com/DavidXu999/Stochastically-Dominant-Peer-Prediction`.

## E.1  Information Structure

For Section 7.2, we consider two real-world crowdsourcing datasets, each having a signal space of size $|\Sigma_1| = 2$ and $|\Sigma_2| = 4$. The information structure between a pair of agents can be characterized by a Dawid-Skene model Dawid and Skene [1979]. In particular, each task has an underlying ground truth $\omega$ that is i.i.d. sampled from a prior distribution $W$. Conditioning on $\omega$, each agent receives a signal $\sigma$ according to conditioned distribution $\Gamma$, where $\Gamma_{i,j} = \Pr[\sigma = j \mid \omega = i]$.

In the first dataset [Baba et al., 2018], 18 agents each labels whether a compound is appropriate or inappropriate to be synthesized. The prior of the ground truth and the conditioned signal distribution are

$$
W_1 = \begin{bmatrix} 0.613 & 0.387 \end{bmatrix}, \qquad \Gamma_1 = \begin{bmatrix} 0.905 & 0.095 \\ 0.283 & 0.717 \end{bmatrix}.
$$

In the second dataset Venanzi et al. [2015], 110 agents provide the annotations of the sentiment of 300 tweets. There are four ground truth labels and four possible signals where the information structure is

$$
w_2 = \begin{bmatrix} 0.196 & 0.241 & 0.247 & 0.316 \end{bmatrix}, \qquad \Gamma_2 = \begin{bmatrix} 0.770 & 0.122 & 0.084 & 0.024 \\ 0.091 & 0.735 & 0.130 & 0.044 \\ 0.033 & 0.062 & 0.866 & 0.039 \\ 0.068 & 0.164 & 0.099 & 0.669 \end{bmatrix}.
$$

## E.2  The Sensitivity of Original Implementations

*How much does the partition rounding reduction harm the sensitivity?* We compare the partition rounding reduction implementations of CA, MA, and PTS with their original implementations in Figure 2. To be clear, other than the implementations considered in Section 7, we consider the following implementation of the mechanisms.

- **PTS** corresponds to the peer truth serum mechanism introduced in Section 4.2, which is truthful but not SD-truthful.

- **CA** corresponds to the original implementation of the Correlated Agreement mechanism described in Section 4.3, which is truthful but not SD-truthful.

- **CA-partition** represents the implementation where we get $K = n/2$ partitions of questions and score agents the average of the individual CA score for each partition (without normalization). By Proposition F.1, CA-partition is SD-truthful when signals are binary.

- **MA** corresponds to the original implementation of the Matching Agreement mechanism described in Appendix D, which is truthful but not SD-truthful.

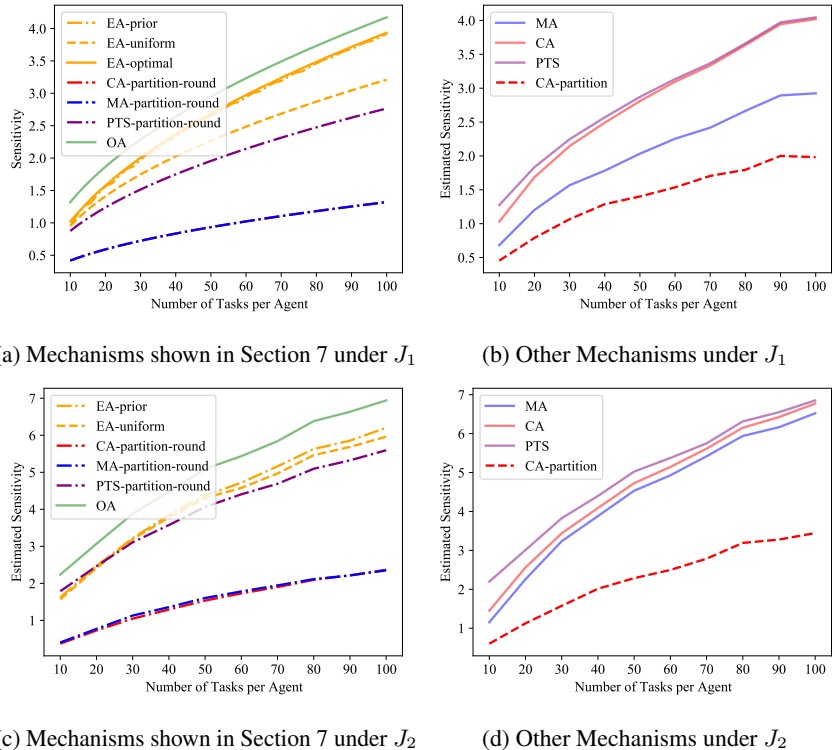

(a) Mechanisms shown in Section 7 under $J_1$      (b) Other Mechanisms under $J_1$

(c) Mechanisms shown in Section 7 under $J_2$      (d) Other Mechanisms under $J_2$

Figure 2: A Comparison of the Sensitivity of Different Mechanisms.

We observe that, relative to the original implementations (solid lines), the partition-rounding reductions (dashed lines) consistently exhibit lower sensitivity. This observation further demonstrates the superior sensitivity of EA mechanisms, as it has the property of SD-truthfulness in the binary setting with no need for partition rounding reduction.

## F    Additional Proofs

### F.1    The CA Mechanism

We first show that a single draw of the CA score is SD-truthful when $|\Sigma| = 2$.

**Proposition F.1.** *When $|\Sigma| = 2$, instead of taking the average, the mechanism that scores agents using an individual CA score $S_i^{CA}$ is SD-truthful.*[4]

*Proof of Proposition F.1.* We first show that the agreement function $T_\Delta$ is an identity matrix if and only if signals are self-predicting. For simplicity, let $J_{\sigma,\sigma'} = \Pr(X_a = \sigma, X_b = \sigma')$ be the joint distribution, and let $M_\sigma^a = \Pr(X_a = \sigma)$ and $M_\sigma^b = \Pr(X_b = \sigma)$ be the marginal distribution of Alice and Bob respectively. The delta matrix can be written as

$$\begin{aligned}
\Delta_{\sigma,\sigma} &= J_{\sigma,\sigma} - M_\sigma^a \, M_\sigma^b \\
&= J_{\sigma,\sigma} - M_\sigma^a \, (J_{\sigma,\sigma} + J_{\sigma',\sigma}) \\
&= J_{\sigma,\sigma} \, M_{\sigma'}^a - J_{\sigma',\sigma} \, M_\sigma^a \\
&= M_\sigma^a \, M_{\sigma'}^a \, \left( \Pr(X_b = \sigma \mid X_a = \sigma) - \Pr(X_b = \sigma \mid X_a = \sigma') \right).
\end{aligned}$$

---

[4]When $|\Sigma| \geq 3$, counter-examples exist showing that scoring agents with $S_i^{CA}$ is no longer SD-truthful for general information structure.

$$\Delta_{\sigma,\sigma'} = J_{\sigma,\sigma'} - M_\sigma^a \, M_{\sigma'}^b$$
$$= J_{\sigma,\sigma'} - M_\sigma^a \, (J_{\sigma,\sigma'} + J_{\sigma',\sigma'})$$
$$= J_{\sigma,\sigma'} \, M_{\sigma'}^a - J_{\sigma',\sigma'} \, M_\sigma^a$$
$$= M_\sigma^a \, M_{\sigma'}^a \, (\Pr(X_b = \sigma' \mid X_a = \sigma) - \Pr(X_b = \sigma' \mid X_a = \sigma')).$$

Next, we reason about the distribution of scores under different strategies. Note that a single draw of the CA score can be $-1$, $0$, or $1$. Therefore, to prove SD-truthfulness for a single draw of the CA score, it is sufficient to show that $\Pr\left(S_i^{CA}(\tau) = 1\right) \geq \Pr\left(S_i^{CA}(\theta) = 1\right)$ and $\Pr\left(S_i^{CA}(\tau) = -1\right) \leq \Pr\left(S_i^{CA}(\theta) = -1\right)$ for any strategy $\theta$. When signals are binary, there are only three types of untruthful strategy: 1) always reporting 0, 2) always reporting 1, and 3) always flipping 0 to 1 and 1 to 0. Any other strategy is a mixture of truth-telling and these strategies. Now, we show that all of these strategies are stochastically dominated by truth-telling by discussing two cases.

When signals are self-predicting, i.e. $\Pr(X_b = \sigma \mid X_a = \sigma) - \Pr(X_b = \sigma \mid X_a = \sigma') > 0$, the distribution of $S_i^{CA}$ is

$$\Pr\left(S_i^{CA}(\tau) = 1\right) = J_{0,0} \, M_1^b + J_{1,1} \, M_0^b,$$
$$\Pr\left(S_i^{CA}(\tau) = -1\right) = J_{0,1} \, M_0^b + J_{1,0} \, M_1^b.$$

Because Bob's signal is self-predicting, $\Pr\left(S_i^{CA}(\tau) = 1\right) > \Pr\left(S_i^{CA}(\tau) = -1\right)$. Let $\theta_f$ be the strategy where Alice always flips the signals.

$$\Pr\left(S_i^{CA}(\theta_f) = 1\right) = J_{0,1} \, M_0^b + J_{1,0} \, M_1^b = \Pr\left(S_i^{CA}(\tau) = -1\right),$$
$$\Pr\left(S_i^{CA}(\theta_f) = -1\right) = J_{0,0} \, M_1^b + J_{1,1} \, M_0^b = \Pr\left(S_i^{CA}(\tau) = 1\right).$$

This means that $S_i^{CA}(\theta_f)$ is dominated by $S_i^{CA}(\tau)$.

Let $\mu$ be the strategy where Alice always reports the same signal.

$$\Pr\left(S_i^{CA}(\mu) = 1\right) = \Pr\left(S_i^{CA}(\mu) = -1\right) = M_0^b \, M_1^b.$$

Therefore,

$$\Pr\left(S_i^{CA}(\tau) = 1\right) - \Pr\left(S_i^{CA}(\mu) = 1\right) = J_{0,0} \, M_1^b + J_{1,1} \, M_0^b - M_0^b \, M_1^b$$
$$= J_{0,0} \, M_1^b + J_{1,1} \, M_0^b - M_0^b \, (J_{0,1} + J_{1,1})$$
$$= J_{0,0} \, M_1^b - J_{0,1} \, M_0^b$$
$$= M_0^b \, M_1^b \, (\Pr(X_a = 0 \mid X_b = 0) - \Pr(X_b = 0 \mid X_a = 1))$$
$$> 0$$

Similarly, we can observe that $\Pr\left(S_i^{CA}(\tau) = -1\right) < \Pr\left(S_i^{CA}(\mu) = -1\right)$.

The analysis for the case when signals are not self-predicting, i.e. $T_\Delta(\sigma, \sigma') = 1$ if and only if $\sigma \neq \sigma'$, is analogous. $\qquad\square$

Next, we show that when $|\Sigma| = 3$, counterexamples exist showing that truth-telling does not FOSD every untruthful strategy.

*Example* F.1. Consider the following joint distribution:

$$J = \begin{pmatrix} 0.02 & 0.01 & 0.02 \\ 0.2 & 0.4 & 0.25 \\ 0.00 & 0.05 & 0.05 \end{pmatrix}.$$

The marginal distribution of Alice and Bob are

$$M^a = (0.05, \quad 0.85, \quad 0.1) \quad \text{and} \quad M^b = (0.22, \quad 0.46, \quad 0.32).$$

The product of marginal distribution and the corresponding agreement function are

$$\Delta = \begin{pmatrix} 0.009 & -0.013 & 0.004 \\ 0.013 & 0.009 & -0.022 \\ -0.022 & 0.004 & 0.018 \end{pmatrix}, \quad T_\Delta = \begin{pmatrix} 1 & 0 & 1 \\ 1 & 1 & 0 \\ 0 & 1 & 1 \end{pmatrix}.$$

When Alice reports truthfully, the probability of obtaining a score of 1, i.e. obtaining a score of 1 on the bonus question and obtaining a score of 0 on the penalty question, is given by

$$\Pr(S_i^{CA}(\tau) = 1) = \underbrace{(J_{0,0} + J_{0,2}) \cdot M_1^b}_{\text{Alice reports 0}} + \underbrace{(J_{1,0} + J_{1,1}) \cdot M_2^b}_{\text{Alice reports 1}} + \underbrace{(J_{2,1} + J_{2,2}) \cdot M_0^b}_{\text{Alice reports 0}} = 0.2324.$$

However, when Alice plays a strategy $\mu$ that always reports 0, the probability of obtaining a score of 1 is given by

$$\Pr(S_i^{CA}(\mu) = 1) = (M_0^b + M_2^b) \cdot M_1^b = 0.2474 > \Pr(S_i^{CA}(\tau) = 1).$$

Therefore, $\mu$ is not first-order stochastically dominated by $\tau$.

## F.2   The Direct Rounding Reduction

*Proof of Proposition 5.2.* By definition, the sensitivity of a mechanism with score $S(e)$ at effort level $e$ is

$$\delta^{\mathcal{M}}(e) = \frac{\nabla m^{\mathcal{M}}(e)}{\mathrm{std}^{\mathcal{M}}(e)}.$$

The direct rounding of $S^{\mathcal{M}}(e)$, denoted as $\tilde{S}^{\mathcal{M}}(e)$, is a Bernoulli variable with success probability $\mathbb{E}[\lambda^{\mathcal{M}}] = \frac{m^{\mathcal{M}}(e) - S_{\inf}^{\mathcal{M}}}{S_{\sup}^{\mathcal{M}} - S_{\inf}^{\mathcal{M}}}$. Therefore,

$$\nabla \mathbb{E}\left[\tilde{S}^{\mathcal{M}}(e)\right] = \frac{1}{S_{\sup}^{\mathcal{M}} - S_{\inf}^{\mathcal{M}}} \cdot \nabla m^{\mathcal{M}}(e)$$

$$\mathrm{std}\left(\tilde{S}^{\mathcal{M}}(e)\right) = \sqrt{\mathbb{E}[\lambda^{\mathcal{M}}]\left(1 - \mathbb{E}[\lambda^{\mathcal{M}}]\right)} = \frac{1}{S_{\sup}^{\mathcal{M}} - S_{\inf}^{\mathcal{M}}} \cdot \sqrt{\left(m^{\mathcal{M}}(e) - S_{\inf}^{\mathcal{M}}\right)\left(S_{\sup}^{\mathcal{M}} - m^{\mathcal{M}}(e)\right)}$$

$$\Rightarrow \quad \tilde{\delta}^{\mathcal{M}}(e) = \frac{\nabla m^{\mathcal{M}}(e)}{\sqrt{\left(m^{\mathcal{M}}(e) - S_{\inf}^{\mathcal{M}}\right)\left(S_{\sup}^{\mathcal{M}} - m(e)\right)}}. \tag{3}$$

$\square$

## F.3   The Partition Rounding Reduction

*Proof of Lemma 5.3.* Let $S_i(\tau)$ and $S_i(\theta)$ be the random variable of each individual score under truth-telling and strategy $\theta$, respectively. $S_i(\tau)$ (and $S_i(\theta)$) is i.i.d. because questions are assumed to be i.i.d. and strategies are task-independent. We want to show

$$S_i(\tau) \quad \succeq_{\mathrm{FOSD}} \quad S_i(\theta) \quad \Leftrightarrow \quad S(\tau) \quad \succeq_{\mathrm{FOSD}} \quad S(\theta).$$

**Forward Direction.** Suppose $S_i(\tau) \succeq_{\mathrm{FOSD}} S_i(\theta)$ for any $\theta$. By definition, this means for each individual score, we can find a coupling such that whenever $S_i(\theta) = s_i$, $S_i(\tau) = s_i' \geq s_i$. Applying this coupling for every individual score and taking the average gives us a coupling for $S(\tau)$ and $S(\theta)$: whenever $S(\theta) = \sum_{i \in [K]} s_i$, $S(\tau) = \sum_{i \in [K]} s_i' \geq \sum_{i \in [K]} s_i$ in the coupling. Therefore, in this coupling, $\Pr(S(\theta) \leq t) \geq \Pr(S(\tau) \leq t)$ for any $t \in [K \cdot S_{\inf}, K \cdot S_{\sup}]$ where $S_{\inf}$ and $S_{\sup}$ are the infimun and supremum of each $S_i$, respectively. This completes the proof for the sufficiency.

**Reverse Direction.** We prove the contrapositive of the statement: if $S_i(\tau)$ does not first-order stochastic dominates $S_i(\theta)$, then $S(\tau)$ does not first-order stochastic dominates $S(\theta)$. By the failure of FOSD for the individuals, there exists some threshold $t_1$ and (by the i.i.d. assumption) every $i$ such that $F_\tau(t_1) > F_\theta(t_1)$, where $F_\tau$ and $F_\theta$ denote that c.d.f. for $S_i(\tau)$ and $S_i(\theta)$ respectively. Let

$F_\tau^k$ and $F_\theta^k$ be the c.d.f. of the sum of $k$ i.i.d. individual scores, i.e. $\sum_{i\in[k]} S_i(\tau)$ and $\sum_{i\in[k]} S_i(\theta)$ respectively. We want to find a $t_K$ such that $F_\tau^K(t_K) > F_\theta^K(t_K)$. Then, because $S$ is a linear scaling (by $1/K$) of the sum, the inequality transfers directly to the averages.

We prove this via an inductive argument:

- Base case ($K = 1$): by assumption, there exists a $t_1$ such that $F_\tau(t_1) > F_\theta(t_1)$.

- Inductive step: suppose that for $K - 1$ there exists some threshold $t_{K-1}$ such that

$$F_\tau^{K-1}(t_{K-1}) > F_\theta^{K-1}(t_{K-1}).$$

  Now, consider the $K$-fold convolution. Note that the convolution of discrete distributions is a weighted sum of the individual c.d.f..

$$F_\tau^K(t_K) = \sum_s \Pr\left(\sum_{i\in[K-1]} S_i(\tau) = s\right) F_\tau(t_K - s),$$

  and similarly for $F_\theta^K(t_K)$. By the base case and the induction assumption, we can set $t_K = t_{K-1} + t_1$ so that $F_\tau^K(t_K) > F_\theta^K(t_K)$. This means that there exists a threshold $t = t_K/K$ so that

$$\Pr\big(S(\tau) \leq t\big) > \Pr\big(S(\theta) \leq t\big).$$

This completes the proof of the necessity. $\qquad\square$

## F.4 EA in the Binary-signal Setting

*Proof of Theorem 6.1.* Suppose Alice observes $m_0$ zeros and $m_1$ ones on $n$ questions while the enforced empirical distribution is $\Phi = (n_0, n_1)$. Alice's strategy and $\Phi$ thus determines the number of i.i.d. samples from each of the four binomials.

To prove SDT, we have to show that truth-telling results in a score distribution that first-order stochastic dominates the score distribution under any other strategy. Suppose W.L.O.G. that $m_0 > n_0$ and the mechanism will randomly flip Alice's reports on $m_0 - n_0$ questions from 0 to 1. If Alice reports truthfully, there are $n_0$ questions where Alice reports 0 and will be scored 1 if Bob also reports 0; there are $m_1$ questions where Alice reports 1 and will be scored 1 if Bob also reports 1; and there are $m_0 - n_0$ questions where Alice observes 0 but her reports are flipped to 1 which means that she will be scored 1 if Bob reports 1 on those questions. Therefore, the final score of Alice can be expressed as

$$n \cdot S(\tau) \sim \mathrm{Bin}(n_0, p_{00}) + \mathrm{Bin}(m_1, p_{11}) + \mathrm{Bin}(m_0 - n_0, p_{01}). \tag{4}$$

To show that $S(\tau)$ dominates $S(\theta)$ for any strategy $\theta$, it is sufficient to focus on deterministic strategies. Suppose $x_{i,j}$ is the number of questions that Alice observes $i$ but reports $j$ for $i, j \in \{0, 1\}$. First note that $x_{0,0} + x_{0,1} = m_0$ and $x_{1,0} + x_{1,1} = m_1$. Then, we argue that it is sufficient to focus on strategies that satisfy $x_{0,0} + x_{1,0} = n_0$ and $x_{0,1} + x_{1,1} = n_1$. Otherwise, the mechanism will enforce $\Phi$ on behalf of Alice which is identical for Alice to flip her signals by herself. Fixing any $\Phi$ and $\Phi_{X_a}$, the above four constraints suggest that there is only one free variable in $x$, meaning that there is only one possible way to cheat—altering $x_{0,0} \in \{0, \dots, n_0\}$.

Now, we show that reporting $x_{0,0} = n_0$ is the best strategy—reporting any $x_{0,0}$ results in a score distribution dominated by reporting $x_{0,0} = n_0$. Also, note that truth-telling is one of the strategies that has $x_{0,0} = n_0$. Let $x_{0,0} = n_0 - k$ where $k \geq 0$. Then, the final score of Alice and be expressed as

$$n \cdot S(\theta) \sim \mathrm{Bin}(n_0 - k, p_{00}) + \mathrm{Bin}(m_1 - k, p_{11}) + \mathrm{Bin}(m_0 - n_0 + k, p_{01}) + \mathrm{Bin}(k, p_{10}). \tag{5}$$

Compared with the truth-telling score distribution in Equation (2), any untruthful reporting can be viewed as subtracting $k$ samples from $\mathrm{Bin}(n_0, p_{00})$ and draw $k$ more samples from $\mathrm{Bin}(k, p_{10})$; subtracting $k$ samples from $\mathrm{Bin}(m_1, p_{11})$ and draw $k$ more samples from $\mathrm{Bin}(m_0 - n_0 + k, p_{01})$.

When signals are self-predicting, $p_{00} > p_{10}$ and $p_{11} > p_{01}$. We can construct a coupling to show that $S(\theta)$ is dominated by $S(\tau)$. In particular, for the $k$ questions whose scores are sampled from $\text{Bin}(k, p_{10})$, we couple them with $k$ out of the $n_0$ samples from $S(\theta) \sim \text{Bin}(n_0 - k, p_{00})$; for the $n_0 - k$ questions whose scores are sampled from $\sim \text{Bin}(n_0 - k, p_{00})$, we couple them with the remaining $n_0 - k$ samples from $\sim \text{Bin}(n_0 - k, p_{00})$; we do the same for the other two Binomial distributions. Under this coupling, it is clear that making $x_{0,0}$ as large as possible will have the dominating score distribution, which happens while truth-telling. $\qquad\square$

### F.5 EA in the Three-signal Setting

*Proof of Proposition C.2.*

**Sufficiency.** We prove the sufficiency by discussing two cases. First, suppose W.L.O.G. that $m_0 \geq n_0$, $m_1 \leq n_1$, and $m_2 \leq n_2$. Under the minimal enforcement rule, the manipulation matrix corresponds to truth-telling is

$$\vartheta^\tau(\rho^*) = \begin{bmatrix} n_0 & n_1 - m_1 & n_2 - m_2 \\ 0 & m_1 & 0 \\ 0 & 0 & m_2 \end{bmatrix}.$$

When we score an arbitrary report vector $\hat{X}_a$, it is sufficient to consider its manipulation matrix $\vartheta$. Therefore, we want to show that $\vartheta^\tau$ will result in a score distribution that first-order stochastically dominates the score distribution of any other feasible $\vartheta$. Note that there are 9 variables in a $3 \times 3$ matrix and 5 constraints: the sum of variables in row $i$ is $m_i$ and the sum of variables in column $j$ is $n_j$ while only 5 of these 6 constraints are independent. This means that any manipulation matrix be characterized by 4 independent variables denoted as $t_1, \ldots t_4 \geq 0$.

$$\vartheta = \begin{bmatrix} n_0 - t_1 - t_2 & n_1 - m_1 + t_1 + t_4 - t_3 & n_2 - m_2 + t_2 + t_3 - t_4 \\ t_1 & m_1 - t_1 - t_4 & t_4 \\ t_2 & t_3 & m_2 - t_2 - t_3 \end{bmatrix}.$$

The variables $t_1, \ldots t_4$ must satisfy the constraints ensuring that every entry in $\vartheta$ remains non-negative.

The final score of truth-telling is given by the average of $n$ Bernoulli variables where $\vartheta^\tau(\rho^*)_{i,j}$ of these variables are sampled from $\text{Bernoulli}(p_{i,j})$. Similarly, the final score corresponding to the manipulation matrix $\vartheta$ is the average of $n$ Bernoulli variables where $\vartheta^\theta_{i,j}$ of them are sampled from $\text{Bernoulli}(p_{i,j})$.

Our goal is to construct a coupling of these Bernoulli variables such that, for any given $\Phi_{X_a}$, $\Phi$, and $t_1, \ldots, t_4$, each variable under truth-telling with manipulation matrix $\vartheta^\tau(\rho^*)$ has a weakly higher probability of success than its coupled counterpart under $\vartheta$. We denote this coupling by $Z^\tau_{i,j} \xrightarrow{k} Z^\theta_{i',j'}$ to indicate that we pair $k$ Bernoulli variables associated with $\vartheta^\tau(\rho^*)$—each drawn from $\text{Bernoulli}(p_{ij})$—with $k$ Bernoulli variables associated with $\vartheta$—each drawn from $\text{Bernoulli}(p_{i'j'})$. Our coupling is:

$$\begin{bmatrix} Z^\tau_{0,0} \xrightarrow{n_0 - t_1 - t_2} Z^\theta_{0,0} & Z^\tau_{1,1} \xrightarrow{t_1 + t_4} Z^\theta_{0,1} & Z^\tau_{2,2} \xrightarrow{t_2 + t_3} Z^\theta_{0,2} \\ Z^\tau_{0,0} \xrightarrow{t_1} Z^\theta_{1,0} & Z^\tau_{1,1} \xrightarrow{m_1 - t_1 - t_4} Z^\theta_{1,1} & Z^\tau_{0,2} \xrightarrow{t_4} Z^\theta_{1,2} \\ Z^\tau_{0,0} \xrightarrow{t_2} Z^\theta_{2,0} & Z^\tau_{0,1} \xrightarrow{t_3} Z^\theta_{2,1} & Z^\tau_{2,2} \xrightarrow{m_2 - t_2 - t_3} Z^\theta_{2,2} \end{bmatrix}.$$

Under this coupling, every diagonal variable $Z^\theta_{i,i}$ is coupled with $Z^\tau_{i,i}$ which is drawn from the same Bernoulli distribution. Furthermore, every non diagonal variable $Z^\theta_{i,j}$ is either coupled with $Z^\tau_{i,i}$ which has a larger success probability because of self-prediction, or coupled with $Z^\tau_{k,j}$ which has the same success probability because signals are uniformly self-predicting.

The proof of the second case is analogous. suppose W.L.O.G. that $m_0 > n_0$, $m_1 > n_1$, and $m_2 < n_2$. Under the minimal enforcement rule, the manipulation matrix corresponds to truth-telling is

$$\vartheta^\tau = \begin{bmatrix} n_0 & 0 & m_0 - n_0 \\ 0 & n_1 & m_1 - n_1 \\ 0 & 0 & m_2 \end{bmatrix}. \tag{6}$$

Any feasible manipulation matrix can be written as

$$\vartheta^\theta = \begin{bmatrix} n_0 - t_1 - t_2 & t_4 & m_0 - n_0 + t_1 + t_2 - t_4 \\ t_1 & n_1 - t_3 - t_4 & m_1 - n_1 + t_3 + t_4 - t_1 \\ t_2 & t_3 & m_2 - t_2 - t_3 \end{bmatrix}. \tag{7}$$

We construct the following coupling. If $t_1 \geq t_4$, we apply the term in red; while if $t_1 < t_4$, we apply the term in blue.

$$\begin{bmatrix} Z_{0,0}^\tau \xrightarrow{n_0-t_1-t_2} Z_{0,0}^\theta & Z_{1,1}^\tau \xrightarrow{t_4} Z_{0,1}^\theta & Z_{2,2}^\tau \xrightarrow{t_2} Z_{0,2}^\theta, \; Z_{1,2}^\tau \xrightarrow{t_1-t_4} Z_{0,2}^\theta \\ Z_{0,0}^\tau \xrightarrow{t_1} Z_{1,0}^\theta & Z_{1,1}^\tau \xrightarrow{n_1-t_3-t_4} Z_{1,1}^\theta & Z_{2,2}^\tau \xrightarrow{t_3} Z_{1,2}^\theta, \; Z_{0,2}^\tau \xrightarrow{t_4-t_1} Z_{1,2}^\theta \\ Z_{0,0}^\tau \xrightarrow{t_2} Z_{2,0}^\theta & Z_{1,1}^\tau \xrightarrow{t_3} Z_{2,1}^\theta & Z_{2,2}^\tau \xrightarrow{m_2-t_2-t_3} Z_{2,2}^\theta \end{bmatrix}.$$

Under the same argument, we can observe that every $Z_{i,j}^\theta$ is coupled with a $Z_{i',j'}^\tau$ with a weakly larger success probability. This completes the proof of sufficiency.

**Necessity.** We first show that if the mechanism's enforcement rule is not minimal, it is not SD-truthful. For a non-minimal enforcement rule, there must exist a report vector $\hat{X}_a$ and an enforced histogram $\Phi$ such that if Alice reports $\hat{X}_a$ truthfully, there exists a diagonal value in the corresponding manipulation matrix that can be increased without breaking the constraints. In other words, there exists a strategy $\theta$ whose corresponding manipulation matrix is $\vartheta^\theta$ and a column $j$ such that $\vartheta_{j,j}^\theta > \vartheta_{j,j}^\tau$. Then, the strategy $\theta$ is not dominated by truth-telling even when signals are uniformly self-predicting. This is because by playing $\theta$, Alice can move $\vartheta_{j,j}^\theta - \vartheta_{j,j}^\tau$ samples from $\mathrm{Bernoulli}(p_{ij})$ or $\mathrm{Bernoulli}(p_{kj})$ to $\mathrm{Bernoulli}(p_{jj})$ where $i, k \neq j$. For self-predicting signals, $p_{jj}$ is the largest column value.

Next, we show that if signals are not uniformly self-predicting, the enforced agreement mechanism is not SD-truthful even when the enforcement rule is minimal. We can easily find a counterexample by comparing Equation (6) with Equation (7). Suppose W.L.O.G. that $\vartheta_{0,2}^\tau < \vartheta_{1,2}^\tau$, then Alice can set $t_1 = t_2 = t_3 = 0$ and $t_4 = \max(n_1, m_0 - n_0)$ in which case she can move $t_4$ samples from $\mathrm{Bernoulli}(p_{0,2})$ to $\mathrm{Bernoulli}(p_{1,2})$, resulting a score distribution that dominates truth-telling. This completes the proof of necessity. $\qquad \square$

### F.6 The Optimal Enforcement

*Proof of Lemma C.5.* Suppose Alice observes 0 on $m_0$ out of the $n$ questions and observes 1 on the remaining $n - m_0$ questions. There are two cases.

First, when $m_0 \leq n_0$, the final score of Alice is the average of the following three type of questions:

- $m_0$ questions where Alice observes 0 and receives a score of 1 if Bob also reports 0.

- $n - n_0$ questions where Alice observes 1 and receives a score of 1 if Bob also reports 1.

- $n_0 - m_0$ questions where Alice observes 1 but receives a score of 1 if Bob reports 0.

When Alice exerts full effort ($e = 1$), Bob's reports (which match his signals) follow the conditional probability distribution $p_{ij} = \Pr(X_b = j \mid X_a = i)$. In contrast, when Alice exerts no effort ($e = 0$), her signals are uninformative about Bob's signals, and thus her belief about Bob's reports follows the marginal distribution $M_i^b = \Pr(X_b = i)$. Therefore, conditioned on Alice observing 0 in $m_0$ questions, the score defined by EA follows the sum of three binomials:

$$n \cdot S^{EA}(n_0, e = 1 \mid m_0 \leq n_0) \sim \mathrm{Bin}(m_0, p_{00}) + \mathrm{Bin}(n - n_0, p_{11}) + \mathrm{Bin}(n_0 - m_0, p_{10}),$$

$$n \cdot S^{EA}(n_0, e = 0 \mid m_0 \leq n_0) \sim \mathrm{Bin}(m_0, M_0^b) + \mathrm{Bin}(n - n_0, M_1^b) + \mathrm{Bin}(n_0 - m_0, M_0^b).$$

Note that for any question, the probability of Alice observing 0 while exerting effort is the marginal distribution $\Pr(X_a = 0)$, which is identical to the probability of observing 0 while exerting no effort. This implies that under our definition of effort, the probability of Alice observing 0 on $m_0$ questions does not depend on the effort. Let $\rho_0 = \sum_{j \in [n]} (1 - X_{a,j})$ be the number of questions where Alice observes 0. The probability that Alice observes 0 on $m_0$ questions is

$$\Pr(\rho_0 = m_0) = \binom{n}{m_0} (M_0^a)^{m_0} (1 - M_0^a)^{n - m_0}.$$

Consequently, when $m_0 \leq n_0$, Alice's score, conditioned on observing 0 in $m_0 \leq n_0$ questions (which occurs with probability $\Pr(\rho_0 = m_0)$), follows the distribution:

$$n \cdot S^{EA}(n_0, e \mid m_0 \leq n_0) \sim \text{Bin}\left(m_0, e\, p_{00} + (1 - e)\, M_0^b\right) + \text{Bin}\left(n - n_0, e\, p_{11} + (1 - e)\, M_1^b\right)$$
$$+ \text{Bin}\left(n_0 - m_0, e\, p_{10} + (1 - e)\, M_0^b\right).$$

In the second case, where $m_0 > n_0$, the final score of Alice is the average of the following three type of questions:

- $n_0$ questions where Alice observes 0 and receives a score of 1 if Bob also reports 0.

- $n - m_0$ questions where Alice observes 1 and receives a score of 1 if Bob also reports 1.

- $m_0 - n_0$ questions where Alice observes 0 but receives a score of 1 if Bob reports 1.

By the analogous arguments, we know that Alice's score, conditioned on observing 0 in $m_0 > n_0$ questions, follows the distribution:

$$n \cdot S^{EA}(n_0, e \mid m_0 > n_0) \sim \text{Bin}\left(n_0, e\, p_{00} + (1 - e)\, M_0^b\right) + \text{Bin}\left(n - m_0, e\, p_{11} + (1 - e)\, M_1^b\right)$$
$$+ \text{Bin}\left(m_0 - n_0, e\, p_{01} + (1 - e)\, M_1^b\right).$$

Putting everything together,

$$S^{EA}(n_0, e) \sim \sum_{m_0=0}^{n_0} \Pr(\rho_0 = m_0) \cdot S^{EA}(n_0, e \mid m_0 \leq n_0) + \sum_{m_0=n_0+1}^{n} \Pr(\rho_0 = m_0) \cdot S^{EA}(n_0, e \mid m_0 > n_0)$$
$$(8)$$

Now, we show that the standard deviation of $S^{EA}(n_0, e)$ is independent of $n_0$. For simplicity, let $\eta_{ij} = e\, p_{ij} + (1 - e)\, M_j^b$. Note that $\eta_{i0} + \eta_{i1} = e\,(p_{i0} + p_{i1}) + (1 - e)\,(M_0^b + M_1^b) = 1$. Furthermore, the variance of a binomial with parameters $n, p$ is $np(1 - p)$. When $m_0 \leq n_0$, the variance of the final score is

$$\text{var}\left(n \cdot S^{EA}(n_0, e \mid m_0 \leq n_0)\right) = m_0\, \eta_{00}\,(1 - \eta_{00}) + (n - n_0)\, \eta_{11}\,(1 - \eta_{11}) + (n_0 - m_0)\, \eta_{10}\,(1 - \eta_{10})$$
$$= m_0\, \eta_{00}\,(1 - \eta_{00}) + (n - m_0)\, \eta_{11}\,(1 - \eta_{11}).$$

Similarly, when $m_0 > n_0$,

$$\text{var}\left(n \cdot S^{EA}(n_0, e \mid m_0 > n_0)\right) = n_0\, \eta_{00}\,(1 - \eta_{00}) + (n - m_0)\, \eta_{11}\,(1 - \eta_{11}) + (m_0 - n_0)\, \eta_{01}\,(1 - \eta_{01})$$
$$= m_0\, \eta_{00}\,(1 - \eta_{00}) + (n - m_0)\, \eta_{11}\,(1 - \eta_{11}).$$

Therefore, conditioned on observing 0 on $m_0$ questions, the standard deviation $\text{std}\left(S^{EA}(n_0, e \mid m_0)\right) = \sqrt{m_0\, \eta_{00}\,(1 - \eta_{00}) + (n - m_0)\, \eta_{11}\,(1 - \eta_{11})}$ is independent of $n_0$. Consequently, the standard deviation of the final score, averaging over $m_0$, is also independent of $n_0$. $\qquad\square$

*Proof of Proposition C.6.* By Lemma C.5, the standard deviation of the EA mechanism is independent of the enforcement $\Phi$. Therefore, the enforcement that maximizes the sensitivity is the one that maximizes the derivative of the expected score w.r.t. effort $e$. By Equation (8), the score given by EA can be expressed as the average of many binomial variables. Note that the expectation of a binomial variable with parameter $(n, p)$ is $n \cdot p$. We thus have

$$n \cdot \mathbb{E}\left[S^{EA}(n_0, e \mid m_0 \leq n_0)\right] = m_0 \eta_{00} + (n - n_0)\, \eta_{11} + (n_0 - m_0)\, \eta_{10}$$
$$n \cdot \mathbb{E}\left[S^{EA}(n_0, e \mid m_0 > n_0)\right] = n_0 \eta_{00} + (n - m_0)\, \eta_{11} + (m_0 - n_0)\, \eta_{01},$$

where $\eta_{ij} = e\, p_{ij} + (1 - e)\, M_j^b$. The derivative of $\eta_{ij}$ w.r.t. $e$ is thus $\eta_{ij}' = p_{ij} - M_j^b$.

We aim to find the $n_0$ that maximizes

$$\frac{\partial \mathbb{E}\left[S^{EA}(n_0, e)\right]}{\partial e} = \sum_{m_0=0}^{n_0} \Pr(\rho_0 = m_0) \cdot \left(m_0\, \eta_{00}' + (n - n_0)\, \eta_{11}' + (n_0 - m_0)\, \eta_{10}'\right)$$

$$+ \sum_{m_0=n_0+1}^{n} \Pr(\rho_0 = m_0) \cdot \left(n_0\, \eta_{00}' + (n - m_0)\, \eta_{11}' + (m_0 - n_0)\, \eta_{01}'\right).$$

Let $n_0$ increase by 1:

$$\frac{\partial \mathbb{E}\left[S^{EA}(n_0 + 1, e)\right]}{\partial e} = \sum_{m_0=0}^{n_0+1} \Pr\left(\rho_0 = m_0\right) \cdot \left(m_0\, \eta'_{00} + (n - n_0 - 1)\, \eta'_{11} + (n_0 + 1 - m_0)\, \eta'_{10}\right)$$

$$+ \sum_{m_0=n_0+2}^{n} \Pr\left(\rho_0 = m_0\right) \cdot \left((n_0 + 1)\, \eta'_{00} + (n - m_0)\, \eta'_{11} + (m_0 - n_0 - 1)\, \eta'_{01}\right).$$

Therefore, the marginal return of increasing $n_0$ is given by

$$d\nabla S^{EA}(n_0) = \frac{\partial \mathbb{E}\left[S^{EA}(n_0 + 1, e)\right]}{\partial e} - \frac{\partial \mathbb{E}\left[S^{EA}(n_0, e)\right]}{\partial e}$$

$$= \sum_{m_0=0}^{n_0} \Pr\left(\rho_0 = m_0\right) \cdot \left(\eta'_{10} - \eta'_{11}\right) + \sum_{m_0=n_0+1}^{n} \Pr\left(\rho_0 = m_0\right) \cdot \left(\eta'_{00} - \eta'_{01}\right).$$

Note that

$$\begin{aligned}
\eta'_{10} - \eta'_{11} &= p_{10} - M_0^b - p_{11} + M_1^b \\
&= -2(p_{11} - M_1^b) && (p_{01} = 1 - p_{11}, \text{ and } M_0^b = 1 - M_1^b.) \\
&= -2(p_{11} - (M_0^a\, p_{01} + M_1^a\, p_{11})) \\
&= -2(p_{11}\,(1 - M_1^a) - p_{01} M_0^a) \\
&= -2\, M_0^a\,(p_{11} - p_{01}) \\
&< 0 && (\text{By self-prediction.})
\end{aligned}$$

Similarly, we can show that $\eta'_{00} - \eta'_{01} > 0$.

Therefore, when $n_0 = 0$, $d\nabla S^{EA}(0) > 0$, and when $n_0 = n$, $d\nabla S^{EA}(n) < 0$. Furthermore, $d\nabla S^{EA}(n_0)$ decreases in $n_0$. The sensitivity-maximizing $n_0$ is thus the largest integer such that $d\nabla S^{EA}(n_0) > 0$. Let $F(k) = \sum_{m_0=0}^{k} \Pr\left(\rho_0 = m_0\right)$.

$$\begin{aligned}
& d\nabla S^{EA}(n_0) > 0 \\
\Leftrightarrow\quad & F(n_0)\left(\eta'_{10} - \eta'_{11}\right) + (1 - F(n_0))\left(\eta'_{00} - \eta'_{01}\right) > 0 \\
\Leftrightarrow\quad & F(n_0) < \frac{\eta'_{00} - \eta'_{01}}{\eta'_{00} - \eta'_{01} + \eta'_{11} - \eta'_{10}}.
\end{aligned}$$

$\square$

