# OpenReview forum: "Stochastically Dominant Peer Prediction"
_NeurIPS.cc/2025/Conference — NeurIPS 2025 poster_

### Official Review · Reviewer_BNRF · 2025-06-07

**Clarity:** 3
**Significance:** 3
**Originality:** 3
**Rating:** 4
**Confidence:** 3

**Summary:**

This paper proposes Enforcement Agreement, a new mechanism for the peer prediction problem that satisfies the novel SD-truthfulness criteria. Whereas other mechanisms rely on linear utility functions in order to satisfy Truthfulness, SD-truthful mecahnisms incentivize truth-telling for general increasing utility functions. It is shown that partition rounding can translate any truthful mechanism with bounded scores into a SD-truthful one (Prop 4.4). Furthermore, EA is shown to be SD-truthful for the binary setting when signals are self-predicting (Thm 5.1), and demonstrated to have high sensitivity. Empirical results compare EA to other mechanisms, including OA, PTS, CA, and MA.

**Questions:**

- The K partitions seems to be analogous to Bootstrapping. Do you need the partition assumption in order to get your results -- instead of, i.i.d sampling with replacement?
- Is line 122-124 the only explanation for why your mechanism can handle monotonic but non-linear utility functions? Also, it's not clear what a utility function is, relative to what is discussed in Section 2.
- Why are single task mechanisms different than multi-task mechanisms? Line 443 in Appendix A refers to Prelec (2004) but doesn't suggest why ideas from that literature doesn't seem to translate to the problem addressed in this paper.

**Ethical Concerns:**

["NO or VERY MINOR ethics concerns only"]

**Final Justification:**

The authors sufficiently addressed my concerns in their rebuttal. I will therefore keep my score.

**Limitations:**

The main result (Thm 5.1) is limited to binary and self-predicating signals. The authors demonstrate that SD-truthfulness does not hold for 3+ elements. I would really like a better understanding in this paper why self-predicting (Def 3.3) is a useful or important concept with respect to either PTS (in Sec 3.2) or in the proof of Thm 5.1. This is introduced as a sufficient condition, but no context is provided for why this condition is important, or in what types of real-world decision problems this condition holds.

**Paper Formatting Concerns:**

(Minor) Is colored text allowed in NeurIPS?

**Quality:**

3

**Strengths And Weaknesses:**

The paper does a good job at introducing the peer prediction problem, explaining how the new proposed concepts (SD-truthfulness, partial rounding, and EA) compare to prior literature (OA, PTS, and CA), and demonstrating why EA satisfies the SD-truthful criteria. The authors do well explaining how and why EA (which incorporates OA with partial rounding) achieves a much better sensitivity ratio than OA with direct rounding. An empirical study compares the results. Overall I found the paper clear and convincing. While I did not check the proofs in the appendix thoroughly, they appear generally correct.

Still, peer prediction is a complicated concept and the paper could use several edits to clean up the presentation. Here are some major comments:
- Can you explain the assumption about Bob's reports being truthful on Line 109? You introduce this relationship with Bayes Nash Equilibrium, which makes sense, but don't refer back to it throughout the paper. It's not clear why this assumption, or being in equilibrium, makes sense across these mechanisms.
- The paper could really make use of a running example to demonstrate these concepts. I think your discussion of Direct and Parititon rounding is clear. But the discussion breaks down when you introduce $\Phi$ on line 278. It seems like the discussion in Section 5 is significiantly detached from the model introduced in Section 2, and it's immediately clear to the reader how to build this relationship. The paper loses its relationship to the applications in lines 37-40 very quickly, so I struggle to understand how it could be directly applied.
- Lines 139-143: You introduce two new concepts here -- budget efficiency and measurement integrity. BE is only used in Appendix E.3; MI is not used at all in the paper. I would recommend reframing this paragraph to suggest that these are "useful to know" concepts related to sensitivity, as opposed to highlighting them as important for the reader to remember later on.
- Although you demonstrate EA has better sensitivity, due to partial rounding, you don't explain this in context. It's not clear how good EA is on this measure dimension.
- The discussion on Lines 304-308 could use significant elaboration in the main text, since they appear to be significant parts of your paper's contribution.
- It's not clear: (1) what $\Phi$ is relative to the model introduced in Section 2, (2) what "information structure" means, and (3) how the joint distribution $Pr(X_a, X_b)$ affects your results -- or if this is just a modeling necessity.


Others:
- Fix \citet -> \citep citation format on lines 26, 38, 54; also in Appendix A
- Line 29: fix to "subjective assessment tasks." Otherwise subjectivity is in contrast with "ground truth"
- Not sure if colored variables in Line 298-299 are acceptable for NeurIPS. Please double check.
- Lines 62-64 high-level description of "sensitivity" could be improved.
- Line 78 -- please explain "enforced count" for readers not familiar with the peer prediction literature
- Line 88 -- why is assessing budget efficiency a "robustness check"?
- Line 130: is $X_a$ the random variable over $\Sigma$? So that if $e=1$ then this reduces to the model discussed in Sec 2.1?
- Citation for Prop 3.2? Is there a corresponding necessary condition for OA to be truthful?
- Line 198 - clarify "depends on implementation": either say "not necessarily" or also mention which implementation would be SD-truthful. Does this contradict Table 1? Are lines 210-211 important enough to be bolded?
- Line 227 - how is sensitivity related to practicality? Since this concept is important throughout the paper, it may be worthwhile on elaborating some of the intuitive findings from [Zhang and Schoenebeck, 2023a] on lines 136-138 (not just the technical relationships).
- Line 334: please include a one sentence description of EA-optimal, and then point to the appendix for the full description
- Inconsistent reference style in bibliography

---

> ### Author Rebuttal · Authors · 2025-07-31
>
> We apologize for several conceptual confusions and will update our paper accordingly for clarification. Below, we provide responses to each of your questions. Please follow up for further clarification if there is any misunderstanding of your questions.
>
> ### **Bayesian Nash Equilibrium**
> The score of a peer prediction mechanism depends not only on the agent being scored (Alice) but also on her peer (Bob). Therefore, our considered truthful guarantee is based on the equilibrium concept. By the definition of an equilibrium, we must show that if everyone else is reporting truthfully, Alice is better off reporting truthfully as well. This leads to the assumption that Bob is always reporting truthfully, which applies to every peer prediction mechanism discussed in the paper. This is a common assumption in peer prediction literature.
>
> ### **EA for the motivating example**
>
> The example we introduced in lines 37-40 is primarily used to motivate the concept of SD-truthfulness, i.e, we want mechanisms whose score distribution under truth-telling FOSD that under any untruthful strategy. The mechanism (EA) introduced in Section 5 is a new way to obtain SD-truthfulness, and thus is a new solution to the grading example in lines 37-40.
>
> To apply EA, suppose a grader scores "+" on 8 out of the 10 assigned submissions, but the instructor knows that only the top 30% of submissions should be "+" and 70% should be "-". Then, by EA, the instructor randomly flips 5 submissions graded "+" by the grader to "-". The grader will be scored based on the probability that her grades agree with another grader on the same 10 submissions (after flipping). To incentivize graders, we then map this score into the letter grades using any step function or any rank-order function. For example, assign A to the 30% students with the highest score; assign B to the top 80%; assign C to the remaining students. EA guarantees truth-telling as long as graders prefer A over B over C, and it does not require partition rounding.
>
> Section 2 introduces the general model for how agents’ signals are correlated, how they can manipulate, and the input/output of a peer prediction mechanism. The general model also applies to section 5.
>
> ### **Budget efficiency and measurement integrity**
>
> These two concepts are indeed only used to motivate why sensitivity is a good metric to look at. We will clarify this.
>
> ### **EA under partition rounding**
> Please clarify if we misunderstood. In addition to sensitivity, in Figures 3 and 4, we show that EA can indeed incentivize a desired effort level at a lower cost of payment compared with other lower-sensitivity mechanisms. EA does not require partition rounding to be SD-truthful.
>
> ### **Other questions**
> * **$\Phi$** is a unique concept for EA, which is the distribution of reports that EA enforces before computing the score. However, this does not violate the general model introduced in Section 2. To clarify, suppose $\Phi$ enforces a distribution of reporting 50 “yes” and 50 “no” on 100 questions. If Alice reports 40 “yes” and 60 “no”, EA will randomly flip 10 “no” answers to “yes” so as to enforce $\Phi$.
> * **Information structure** refers to the joint distribution of agents’ signals. We will try to use the latter more in the paper.
> * The **joint distribution** does not affect the truthful guarantees, but it affects the sensitivity of mechanisms. The sensitivity is higher when agents’ signals are more correlated.
> * **Colored variables**: We do not see any regulations about colored text on NeurIPS website. We will remove color edits for the potential camera-ready.
> * Line 78: See our response to “Motivating example”.
> * Line 88: We consider the budgetary efficiency as a **robustness check** because it is shown in the previous paper [Zhang and Schoenebeck 2023] that when the score distribution is approximately Gaussian, a mechanism with a higher sensitivity has higher budgetary efficiency for rank-order utility functions. We replicate the experiments here to show that EA, which has the highest sensitivity, indeed has the highest budgetary efficiency (lowest cost of payments) in our setting.
> * Line 130: Yes, when $e=1$, the model reduces to the general model in Section 2. To further clarify the **effort model**, when both agents exert effort, their signals are drawn from the joint distribution $\Pr(X_a, X_b)$; if either agent does not exert effort, the signals are drawn independently from the marginals $\Pr(X_a)*\Pr(X_b)$. Thus, a higher effort level $e$ corresponds to a stronger correlation between the agents’ signals.
> * One **citation for Proposition 3.2** is [Faltings et al., 2017] (Theorem 1). The necessary condition—OA is not truthful for self-predicting signals is presented in Proposition 3 of [Faltings et al., 2017]. We will include this necessary condition in our paper.
> * Line 198: The original implementation of CA is not SD-truthful, because of the bolded text in line 210. We bold this sentence as this is an interesting insight and an important note for future research. The partition rounding implementation of CA is SD-truthful. The captain of the table clarifies that the table is referring to the rounded version of certain mechanisms.
> * The technical **importance of sensitivity** has been highlighted in several prior works [Zhang and Schoenebeck (2023); Xu et al. (2024); Burrell and Schoenebeck (2023)]. At a high level, these works suggest that peer prediction mechanisms with high sensitivity are better able to produce scores that reflect high effort and high-quality responses. We will include more details about the conditions and technical meanings for these results in a new supplementary section.
>   * In addition to this technical importance, we provide an alternative way to interpret sensitivity. To ensure truthfulness, many peer prediction mechanisms—such as PTS—reweight scores, or—like CA and EA—inject noise. These modifications inevitably reduce the mechanism’s ability to capture and reward the informational content in agents’ responses. In contrast, a high-sensitivity mechanism requires less distortion to ensure truthfulness, preserving more signal and thus providing scores that better reflect the quality of input.
> * Line 334: Feel free to further clarify the question? In the current version, we are already doing what you asked: have one sentence describing the idea and defer details to the appendix. However we will make this clearer.
>
> ### **Partition round with replacement**
>
> As highlighted in line 210, repeatedly using the same question to compute multiple scores will distort the incentive for truth-telling. Although we don’t have a counterexample for every peer prediction mechanism, the intuition from Example 3.1 strongly warns against reusing questions.
>
> ### **Utility function**
> Lines 122-124 are sufficient to show the equivalence between SD-truthfulness (in scores) and truthfulness for any monotone utilities. This follows directly from the definition of first-order stochastic dominance. A utility function maps from a score to a real-value utility, which is a numerical representation of an individual's preferences over a set of outcomes. It is assumed that agents’ objective is to maximize their utilities.
>
> ### **Single-task mechanisms**
> Single-task mechanisms require agents to report both their own answers and probabilistic predictions of their peers' responses. These mechanisms are typically suited for settings where such predictive information is accessible and the questions exhibit a thinking hierarchy—that is, agents who know the correct answers can better anticipate the mistakes of others, but not vice versa. As a result, their use cases and scoring methods differ fundamentally from those of multi-task mechanisms and fall outside the scope of this paper.
>
> ### **Self-prediction**
> Intuitively, self-prediction is a mild assumption in practice. If Alice observes “A”, self-prediction only requires her to believe that Bob is more likely to have also observed “A” than if she had observed a different signal. This excludes only cases when agents' beliefs are very different, such as when they are negatively correlated.
>
> More formally, in Footnote 2, we note that self-prediction is equivalent to positive correlation of agents’ signals in the binary setting—a reasonably weak assumption. Importantly, it is strictly weaker than self-dominance.
>
> We adopt the self-prediction assumption from Faltings et al. (2017), where it is shown to be technically reasonable. Specifically, they demonstrate that Bayesian updating under a Dirichlet prior—a widely used prior over categorical distributions—satisfies the self-prediction condition.
>
> In contrast, self-dominance is much stronger. It requires that upon observing “A”, Alice believes Bob is more likely to observe “A” than any other signal. This can easily be violated in practice. For instance, if “A” is a majority signal with high prior probability and Alice observes a minority signal “B”, she may believe Bob is still more likely to observe “A” than “B”, thereby violating self-dominance.

---

> > ### Comment · Reviewer_BNRF · 2025-08-05
> >
> > Thank you for your comments. I will keep my positive score.

---

> ### Author Response · Authors · 2025-08-06
> **Thank you for your feedback**
>
> Dear reviewer BNRF,
>
> Thank you for your review. We look forward to updating the paper according to your suggestions. Please feel free to follow up with any questions (if any). We'd love to keep the discussion going.
>
> Best,
>
> Authors of Submission23815

---

### Official Review · Reviewer_WpLs · 2025-07-03

**Clarity:** 2
**Significance:** 2
**Originality:** 2
**Rating:** 4
**Confidence:** 4

**Summary:**

The paper introduces a new notion of truthfulness for peer prediction mechanisms called stochastically dominant truthfulness (SD-truthfulness). They then provide several contributions connected to this definition:
(1) They propose a modification of the (known) rounding/normalization strategy that converts an existing peer-prediction mechanism into one that is SD-truthful, such that the resulting modified peer prediction mechanism additionally has higher sensitivity compared to vanilla rounding.
(2) They then propose a novel peer prediction mechanism, called Enforced Agreement (EA), that instead of performing using a rounding strategy proceeds via a different method on top of Output Agreement --- this method provably achieves SD-truthfulness in the binary setting (though not generally/necessarily SD-truthful beyond that), and establish several further properties of EA.
(3) They perform experiments establishing the favorable performance of the SD-truthful methods they have developed, as compared to prior methods with that property --- establishing that SD-truthfulness via their constructions (EA and the "bucketed rounding") leads to better sensitivity curves (i.e. to better sensitivity across various levels of effort).

**Questions:**

See more details in the Weaknesses section, but the following are distilled important questions to resolve:

(1) Please concretely discuss why/how SD-truthfulness would get violated (or not) in complex mechanisms (such as mutual information-based).

(2) Please concretely discuss in substantially more detail why sensitivity is an important metric (currently it's a few lines in the middle of the paper, which is not enough for a key notion), and why one shouldn't worry about defining sensitivity with respect to the utility functions in this case.

(3) Please make general clarity-inducing improvements targeting less experienced readers in terms of extra terminology, context from the elicitation literature, etc.

**Ethical Concerns:**

["NO or VERY MINOR ethics concerns only"]

**Final Justification:**

My conceptual and concrete questions about the setup (the significance of the metric, truthfulness issues in related mechanisms, etc.) were sufficiently clarified and addressed in the response by the authors. Furthermore, I have also studied the other reviews and the authors' rebuttals to those, and believe that the authors have also clarified other reviewers' points to a sufficient extent. The core contributions of the paper are simple and insightful --- however the issue of readability and supporting discussions in the paper will be important but not fully trivial to fix, which is why I kept my (positive) score rather than increase it.

**Limitations:**

Yes.

**Quality:**

3

**Strengths And Weaknesses:**

As the Strengths of this paper, the following stand out:

++ The idea of defining SD-truthfulness is quite simple and neat: due to the canonical link between stochastic dominance and monotonic transformations of random variables, it implies strictly stronger truthfulness guarantees than the linear-utility ones explored before, and thus naturally cover a variety of natural settings where handling a generic monotonic transform of the scores is necessary (such as the given example of students targeting grades A, B, C, and many other conceivable structured settings).

++ The constructions for both the new rounding scheme and for EA are pleasantly simple --- almost to the point of being elegant (being based in particular on clever constructions that permit the use of stochastic coupling to enable meaningful stochastic dominance comparisons), and in that sense are in my opinion a perfect starting point for further, more sophisticated studies/follow-ups that would further explore SD-truthfulness.

++ The empirics do a fairly good job exposing the benefits afforded by the proposed methods, from directly providing sensitivity plots to (in the Appendix) exploring positive consequences on budgeting.

As far as the weaknesses:

-- Why is the sensitivity notion, as defined, the important/the right one to consider? First off, budgeting and fairness that are mentioned existing applications, are a fair point, but they are not elaborated on sufficiently in the main part for a first-time reader's benefit (nor, unfortunately, in the appendix modulo the extra experiments). More fundamentally, I like the sensitivity definition as-is in the standard context of linear utilities (that are modulo offset proportional to scores) --- but since you are considering general monotonic utilities here as your main contribution, a sensitivity notion that would strike me as far more natural would measure e.g. the worst-case (or average case, etc) sensitivity with respect to the utility functions in a certain class of monotonic utilities. In other words, why should I care more about how sensitive the scores themselves are, rather than the utilities generated from those scores? The moment things become nonlinear these are two different notions.

-- One important technical point got swiped under the rug (despite being mentioned twice: in the intro and in the appendix): For a multitude of mechanisms that are more "complex" than the ones discussed in this manuscript, in particular ones based on mutual information etc., it is claimed without further elaboration that due to this complexity, such mechanisms would be "unlikely" to be SD-truthful due to the strategic manipulations that are permitted through these complexities. For the readers' benefit, I believe it would be quite critical to get more concrete on this point. If you believe that a general and simple to state class of manipulations would prevent SD-truthfulness from holding, I'd like to request that you please give such a construction in your response/in the paper. On my end, while I completely agree that exact SD-truthfulness is elusive, in my opinion just citing the complexity of scoring rules as a major reason sounds a bit peculiar/non-fundamental. Moreover, when it comes to looking at approximate rather than exact SD-truthfulness, the logic of saying "the scoring is way too complex, clearly can be exploited" would seem to not be robust to that (as such complexity could perfectly well hide inside the approximation parameter in that case, and possibly in a pretty inconspicuous way).

-- The paper may not be 100% clearly written from the perspective of a general ML/GT reader at times, which is because the paper was written with a specific audience in mind (more so for a generic EC attendee than a generic NeurIPS attendee); in particular, some notions --- like Bayes Nash equilibrium --- get thrown around quite causally throughout the text, and while it's helpful for the seasoned expert reader, I would have like to see further effort on the accessibility front (i.e. either explain such terms at least informally at least in footnotes, or try to avoid them unless necessary). Moreover, even from the experienced reader's perspective there is some lack of clarity in places, see the above Weaknesses.

---

> ### Author Rebuttal · Authors · 2025-07-31
>
> We sincerely thank Reviewer WpLs for their thoughtful and constructive feedback. Your questions highlight important aspects of our work that deserve deeper clarification. We address each of your main concerns below.
>
> ### **SD-truthfulness for Mutual Information Mechanisms**
>
> Thank you for raising this excellent point. We will add a supplementary section in the appendix to provide a more detailed explanation of why mutual information (MI)-based mechanisms are not SD-truthful. Below, we briefly summarize the key ideas.
>
> * First, a straightforward way to implementing MI-based mechanisms is to estimate the empirical joint distribution and score agents using an $f$-divergence between the joint and the product of marginals, i.e. $S^{fMI} = D_f(\Pr(X_a, X_b) || \Pr(X_a)\Pr(X_b))$, as proposed by Kong and Schoenebeck (2017). However, this implementation is only asymptotically truthful. When the number of tasks is small, there is a non-trivial probability that truth-telling is not the best response. For example, if Alice observes "yes" on all her questions, she knows that truth-telling will always result in a score of 0, regardless of Bob's responses (because the estimated joint distribution is the same as the product of marginals). By misreporting "no" on some questions, she introduces variation and has a chance of receiving a positive score. In comparison, such deviations are discouraged under EA because the mechanism will randomly flip the responses on behalf of the agent.
>
> * Several follow-up ideas, such as the Determinant Mutual Information (DMI) mechanism by Kong (2020) and the pairing mechanism by Yu and Schoenebeck (2020), propose various ways to estimate the mutual information. These mechanisms aim to reduce the required number of questions to achieve truthfulness. However, they are still not SD-truthful. Consider the strategy from Example 3.1, where Alice always reports "yes." Under both DMI and the pairing mechanism, this strategy always yields a score of zero, while truth-telling may result in a negative score (we have new empirical results that verified this). Thus, this constant-reporting strategy is not dominated by truth-telling under these mutual information-based mechanisms with negative scores.
>
> In summary, mechanisms with complex score distributions tend to introduce opportunities for strategic manipulations not dominated by truth-telling. We hope these new discussions and results could further reinforce the intuition that SD-truthful mechanisms tend to prefer simple design.
>
> ### **Sensitivity Definition and Motivation**
>
> We appreciate your insightful question about why sensitivity, as defined, is the appropriate metric to consider. Our motivation for considering sensitivity was mainly justified in the previous works, and we acknowledge that we should provide a more detailed discussion in the paper.
> * First, Zhang and Schoenebeck (2023) show that, when the score is normally distributed (which it is when the number of questions is large), sensitivity is a sufficient statistic for budgetary efficiency when agents' utilities depend on the ranking of scores — a mechanism with a higher sensitivity can incentivize the same effort level in equilibrium at a lower cost of payment. Second, Xu et al. (2024) show that there is a one-to-one mapping between sensitivity and measurement integrity [Burrell and Schoenebeck (2023)], meaning that a mechanism with a higher sensitivity can produce scores that are better correlated with the quality of responses.
>
> * A key feature of sensitivity is that it depends only on the mechanism itself, not on the agent's utility function or how scores are translated into rewards. Comparing mechanisms under a fixed utility function is challenging because their score scales differ: for instance, fMI mechanisms produce non-negative scores, DMI scores can be unbounded, while most mechanisms in this paper yield scores between 0 and 1. As a result, each mechanism would require a different (non-linear) transformation of scores to determine rewards, making direct comparison of worst- or average-case sensitivity potentially misleading. A more meaningful comparison can be made under rank-order utility functions, where an agent’s utility depends only on their score ranking. In this setting, Zhang and Schoenebeck (2023) show that sensitivity is the appropriate metric for evaluating mechanisms.
>
> * If we focus on non-linear utilities that are not ranking-based, e.g., a threshold utility function, we should then compare the mechanisms under the mechanism-dependent thresholds optimized for an objective that the principal cares about. For example, suppose the principal aims to elicit a desired effort $e$ at a lower cost, where the marginal cost of effort is $dc$. To incentivize such an effort, the principal must guarantee that agents’ marginal utility of exerting a higher effort overcomes the $dc$, i.e., $r\cdot dp\ge dc$ with $r$ being the payment and $dp$ being the marginal probability of receiving a score that is higher than the threshold. Because $dc$ is fixed, the threshold that minimizes $r$ is the one that maximizes $dp$. Suppose the score is normally distributed, the $dp$-maximizing threshold is precisely the expected score. This means that the original sensitivity is again a sufficient statistic for the performance of peer prediction mechanisms under threshold utilities.
>
> ### **Clarity and Accessibility Improvements**
>
> We greatly appreciate your feedback regarding clarity for a broader audience. From your questions and comments, we can see that you understood the main points and contributions of our work. However, we also recognize that the manuscript could be further improved—for example, by providing a crisp explanation of concepts like Bayes Nash Equilibrium and better contextualizing terminology from the elicitation literature.
>
> We believe that the same improvements that would benefit a larger audience will benefit all readers. You can certainly expect to see changes in the potential camera-ready version that enhance accessibility while maintaining technical rigor. We spent considerable time on the current draft and believe it is suitable for acceptance, but we are committed to making the work more accessible to the broader NeurIPS community.

---

> > ### Comment · Reviewer_WpLs · 2025-08-05
> >
> > Thank you for your response. The additional background you have provided here was indeed quite helpful and worth including in the revised manuscript. Generally on the presentation front, you can see from the discussions with other reviewers that the writing should be improved to avoid potential confusion about various modeling aspects. Subtler points such as contrasting asymptotic and nonasymptotic truthfulness guarantees, as well as the current sensitivity metric being a sufficient statistic beyond the vanilla utilities setting, is also the type of fine print that would further clarify the lay of the land to the readers!
> >
> > I still like the contributions of the paper (modeling and the accompanying theory), and will continue to support it.

---

> ### Author Response · Authors · 2025-08-06
> **Thank you for your feedback**
>
> Dear reviewer WpLs,
>
> Thank you for your appreciation and your thoughtful comments — they've certainly helped sharpen our paper! We're excited to incorporate these clarifications in the next revision. If any other ideas or questions come to mind, please feel free to bring them up. We’d love to keep the conversation going.
>
> Best,
>
> Authors of Submission23815

---

### Official Review · Reviewer_X4e2 · 2025-07-03

**Clarity:** 3
**Significance:** 3
**Originality:** 3
**Rating:** 4
**Confidence:** 3

**Summary:**

The paper considers peer prediction mechanisms for eliciting truthful behavior.

It proposes a new type of truthfulness guarantee, stochastic dominance-truthful (SD-truthful), in which the probability that the mechanism assigns a score above some threshold $t$ to the truth-telling strategy is greater than the probability that it assigns a score above that threshold for any other strategy, for any threshold $t \in \mathbb{R}$.

It also discusses mechanism sensitivity, a measure of the extent to which the mechanism gives higher scores to more informative reports. The goal of the paper is "to design SD-truthful mechanisms that also achieve high sensitivity."

The paper makes the following claims:

1. No existing peer prediction mechanism is naturally SD-truthful under general information structures.
2. Achieving SD-truthfulness is easy in principle but challenging in practice.
3. The proposed _enforcement agreement_ mechanism is more sensitive than the _direct rounding_ or _partition rounding_ reductions, and requires weaker assumptions (signals are self-predicting) for SD-truthfulness than _output agreement_ (signals are self-dominating).

**Questions:**

1. In Figure 1, the paper provides a comparison of the sensitivity of different mechanisms. Is there a way to also evaluate and visualize truthfulness?

2. The legend for Figure 1 (a) and (b) includes CA-partition-round, but I don't see the red line. Is it missing from the plots?

**Ethical Concerns:**

["NO or VERY MINOR ethics concerns only"]

**Final Justification:**

My rating is unchanged. I still think the paper should be accepted, but I also feel it could be written better, to cater to a broader audience. For example, the definition of "effort" still does not make sense to me.

**Limitations:**

See weaknesses above.

**Quality:**

3

**Strengths And Weaknesses:**

### Strengths

- The new SD-truthfulness property is presented clearly.
- The paper does a good job of motivating both SD-truthfulness and sensitivity.
- The paper reviews several prior mechanisms and well situates the new SD-truthfulness property within that context.
    - It shows that SD-truthfulness is only achieved (by output agreement) under the strong assumption of self-dominating signals.
    - It also shows that peer truth serum (PTS) is truthful, but not SD-truthful, under the weaker assumption of self-predicting signals. The justification feels right, although I did not check the details of the proof carefully.
- The new mechanism, enforced agreement (EA) is clearly presented. It is also SD-truthful for binary, self-predicting signals (which is a weaker assumption than OA requires), while being more sensitive under the partition rounding reduction than correlated agreement (CA), matching agreement (MA), and (usually) PTS.

Overall, this feels like a valuable contribution that future work can build on, both for the SD-truthfulness property and for the novel EA mechanism.

### Weaknesses

The proposed mechanism, enforced agreement (EA), is not generally SD-truthful in the non-binary setting. The paper does a good job of pointing out this limitation in the main text, and discusses it in depth in Appendix C. It also points out that EA is still truthful as long as the information structure is known. I am not an expert, so I am not sure what this information structure assumption means or whether it is reasonable.

The explanation of "effort" in the definition of sensitivity was hard for me to follow. I get it at an intuitive level, but the clarity of this section could definitely be improved.

---

> ### Author Rebuttal · Authors · 2025-07-31
>
> We appreciate your acknowledgment of our key contributions and the valuable feedback! We provide our responses below and wish to update our paper for clarification based on your comments.
>
> ### **Binary setting**
>
> While the Enforced Agreement (EA) mechanism is SD-truthful only in the binary-signal setting, we believe it remains valuable for two main reasons.
> * First, the binary-signal setting is a practical and important setting. Many real-world tasks in crowdsourcing naturally take binary form—for example, selecting the better of two responses in RLHF, or determining whether a news article is real or fake in verification tasks. In theory, it is also a widely considered setting. Many peer prediction mechanisms are particularly designed for binary settings, e.g., [Dasgupta and Ghosh (2013)] and [Chen et al. (2024)].
> * Second, EA is exceptionally simple to implement. Unlike PTS, it does not require reweighting scores based on prior probabilities, and unlike CA and related mechanisms, it does not require distinguishing between bonus and penalty questions. Its simplicity is on par with Output Agreement (OA), making it easy to communicate and apply in practice, such as in labeling tasks or educational settings.
>
> We view EA as a simple yet effective step toward addressing the challenge of designing SD-truthful mechanisms. As some reviewers have noted, it opens the door to a promising line of research with substantial opportunities for extension and improvement.
>
> ### **Information structure assumption**
>
> At a high level, the information structure refers to how agents' signals are jointly distributed. To extend EA to multiple signals, we aim to estimate the joint distribution $\Pr(X_a, X_b)$ between agents’ signals. This is not a desired assumption, but also not an unrealistic one. A common approach in peer prediction—used in works such as Dasgupta & Ghosh (2013), Shnayder et al. (2016), and Zhang & Schoenebeck (2023)—is to have each agent answer multiple i.i.d. questions, which allows this joint distribution to be learned empirically. They show that agents are better off helping the designer learn this joint distribution accurately. Therefore, the truthfulness property still holds. This same approach can be readily applied to EA extend to non-binary settings.
>
> ### **Effort model**
> We apologize for the confusion and will clarify this in the paper. Effort is modeled as the probability that an agent chooses to work diligently on a question. When Alice exerts effort (to show that truth-telling is an equilibrium we can assume Bob will always exert effort and report truthfully), the pair of signals from two agents for that question is drawn from the joint distribution $\Pr(X_a, X_b)$; if Alice does not exert effort, the pair of signals is drawn from the marginals $\Pr(X_a)\cdot\Pr(X_b)$. Thus, a higher effort level $e$ corresponds to a stronger correlation between the agents’ signals.
>
>
> ### **Visualize truthfulness**
> Truthfulness is a binary property—a mechanism either satisfies it or it does not. We have formally proven that all mechanisms shown in the visualizations are (SD-)truthful.
>
> The space of all deviations is quite complex, and it is not known how to measure how "close to" truthfulness a mechanism is in a theoretical way.  While [Burrell and Schoenebeck], cited in the paper, create an innovative way to measure truthfulness, it is very empirical and less rigorous.  First, they need to select prior, then they need to select possible deviation strategies. Then they measure whether the agents benefit when the deviating strategies are employed. Our analysis is much stronger than this because we prove that agents cannot benefit from any strategy while other agents are expending effort and telling the truth.
>
> ### **CA-partition-round**
>
> As shown in Proposition D.3, CA and MA have the same sensitivity under the partition rounding reduction in the binary setting. This results in the red lines in the figure being fully overlapped. We apologize for the confusion caused by the removal of several minor results for space considerations and will add a pointer to this clarification in the main body.

---

> > ### Comment · Reviewer_X4e2 · 2025-08-04
> >
> > Thanks for this response. What I meant by question 1 was: is there some way we can be sure it's working from an empirical perspective?

---

> ### Author Response · Authors · 2025-08-04
>
> Thank you for the thoughtful follow-up. We’d like to clarify that the main contribution of our paper is theoretical. While we believe our proposed mechanism offers practical advantages—particularly its simplicity and stronger incentive properties—empirical validation would require additional human-subject experiments, which we view as beyond the scope of the current work and deserving of a separate study.
>
> That said, there is a growing body of empirical research demonstrating the effectiveness of peer prediction mechanisms, even if the exact mechanism we propose has not yet been tested. For instance, Brown et al. (2025), "Incentive Mechanisms for AI: Theory and Evidence," show that a variant of the peer agreement mechanism can successfully incentivize effort while lowering the need for costly auditing. Similarly, Radanovic et al. (2016), "Incentives for Effort in Crowdsourcing Using the Peer Truth Serum," demonstrate that a version of the Peer Truth Serum (PTS) outperforms simpler agreement-based mechanisms in encouraging careful grading.

---

> > ### Comment · Reviewer_X4e2 · 2025-08-08
> >
> > I see, okay. Thanks for the response!

---

### Official Review · Reviewer_AWMx · 2025-07-12

**Clarity:** 3
**Significance:** 3
**Originality:** 3
**Rating:** 4
**Confidence:** 3

**Summary:**

The authors study mechanism design for peer prediction. Whereas theoretically scoring rules that guarantee truthfulness in expectation is sufficient, in practice non-linear payment rules are preferred for incentivizing truthful behavior. To address this discrepancy, the authors introduce a new stronger notion of truthfulness namely stochastically dominant (SD) truthfulness where the truth-telling reward distribution must first-order dominate the reward distributions of all other strategies.

The authors observe that SD-truthfulness does not hold for many mechanisms but can be trivially be attained by rounding all rewards to two extreme values so that the expectation is preserved. This directly yields SD-truthful mechanisms but introduces significantly higher variance in rewards. To address this, they seek to optimize an additional objective, namely sensitivity. They propose an optimized SD-truthful mechanism called Enforced Agreement (EA) that reduces sensitivity and empirically show that its levels of sensitivity closely follow that of other peer-prediction mechanisms while also being SD-truthful.

**Questions:**

- Is it possible to theoretically establish optimal mechanisms that satisfy SD-truthfulness for any domain. Is EA optimal in any regard?
- Is there a characterization for SD-truthful mechanisms?

**Ethical Concerns:**

["NO or VERY MINOR ethics concerns only"]

**Final Justification:**

Based on the rebuttal, the authors give more motivation for the setting considered that they plan to integrate in the paper.

**Limitations:**

The authors clearly state the limitations of their proposed mechanism.

**Quality:**

3

**Strengths And Weaknesses:**

Strengths:
The paper clearly motivates why expected utility truthfulness is insufficient in practice. It introduces SD-truthfulness as a more robust guarantee for arbitrary monotonic preferences. It proves that classical mechanisms (Output Agreement, PTS, Correlated Agreement) do not meet this stronger criterion and designs EA, a novel and creative mechanism to achieve SD-truthfulness in binary domains.

Weaknesses
An important weakness with the proposed mechanism is the lack of theoretical exploration. The paper mainly proposes the EA mechanism as a novel alternative but does not establish its theoretical performance or optimality in some domain. There could be other mechanisms that are better and even not-restricted to give binary outputs.
Moreover, while the proposed mechanism aims to be more practically relevant, its limitation to binary-signals feels substantial in practice.

Finally, the main body of the paper seems very imbalanced. It spends many pages describing the limitations of other mechanisms and spends less than a page to the description of the proposed alternative which very dense, hard to parse and only focused on the SD-truthfulness of EA without motivation.

---

> ### Author Rebuttal · Authors · 2025-07-31
>
> We sincerely thank Reviewer AWMx for their thoughtful review and constructive feedback. We appreciate your recognition of our paper's clear motivation for SD-truthfulness and the novel EA mechanism design. Your feedback helps us identify important areas for improvement and clarification. Below, we clarify several questions and concerns raised in your review.
>
> ### **The Space and Optimization of SD-truthful Mechanisms**
>
> You raise an excellent theoretical question about the possibility of finding optimal SD-truthful mechanisms. Previous efforts to carve out the space of peer prediction mechanisms for optimization have led to limited interest (and success) primarily for two reasons.
> * First, it is too hard to constrain the space of mechanisms conditioned on truth-telling being a BNE. A large issue is that it must work for all prior distributions, but the space of distributions is large. One might overcome this if the joint distribution of agents' signals is known (so that it needs only to work for one prior distribution). However, this is a very strong and limiting assumption. Although this distribution could, in principle, be learned from reported data, doing so introduces two issues: 1) the learned model is often inaccurate and prone to overfitting; 2) since the learned joint distribution depends on reports themselves, agents may have incentives to misreport so as to mislead the optimizer. Previous attempts in this direction have only led to optimization in a very limited space of mechanisms.
> * Second, optimized mechanisms tend to be complex and difficult to interpret. They require the designer to specify, in advance, the scoring or payment rule for every possible signal pair $(X_a, X_b)$. Such a mechanism can hardly be implemented or justified in real-world settings where simplicity and transparency are critical. This may explain why simple mechanisms like Output Agreement (OA), despite their limited theoretical guarantees, remain popular in practice.
>
> To answer your questions, it is theoretically possible to find an SD-mechanism that is more sensitive than EA. However, we doubt it will come from any sort of explicit optimization. Furthermore, even if a more sensitive mechanism were arrived at, it would take a huge advance to show that it was optimal in any meaningful way (or to show EA is optimal for that matter).
>
> ### **Binary setting**
>
> While the Enforced Agreement (EA) mechanism is SD-truthful only in the binary-signal setting, we believe it remains valuable for two main reasons.
> * First, the binary-signal setting is a practical and important setting. Many real-world tasks in crowdsourcing naturally take binary form—for example, selecting the better of two responses in RLHF, or determining whether a news article is real or fake in verification tasks. In theory, it is also a widely considered setting. Many peer prediction mechanisms are particularly designed for binary settings, e.g. [Dasgupta and Ghosh (2013)] and [Chen et al., (2024)].
> * Second, EA is exceptionally simple to implement. Unlike PTS, it does not require reweighting scores based on prior probabilities, and unlike CA and related mechanisms, it does not require distinguishing between bonus and penalty questions. Its simplicity is on par with Output Agreement (OA), making it easy to communicate and apply in practice, such as in labeling tasks or educational settings.
>
> We view EA as a simple yet effective step toward addressing the challenge of designing SD-truthful mechanisms. As some reviewers have noted, it opens the door to a promising line of research with substantial opportunities for extension and improvement.
>
> ### **Addressing the Imbalanced Main Body**
>
> We apologize for not motivating EA clearly enough. We aim to motivate EA throughout Sections 3-4, where we systematically demonstrate why existing mechanisms fail to achieve SD-truthfulness. In particular, Example 3.1 shows that "always reporting the same signal" is a strategy that thwarts the SD-truthfulness of CA. This is because agents can strategically adjust (reduce) the variance of their scores by misreporting, resulting in a score distribution not dominated by truth-telling. Therefore, EA prevents this. It effectively controls the prior distribution of reports so that misreporting cannot change the variance of scores. If an agent's reports do not match the preset distribution, it flips the reports so that it does. Because of this, an agent cannot artificially increase or decrease variance by submitting more or fewer reports of a particular signal.
>
> The current paper structure deliberately builds this motivation: Section 3 establishes that existing mechanisms fail SD-truthfulness, Section 4 shows that simple rounding approaches have limitations, and Section 5 introduces EA as a principled solution. However, we recognize that this connection could be made more explicit. In the potential camera-ready version, we will enhance the presentation to make the logical flow from the limitations of existing approaches to EA's design more transparent to readers.

---

> > ### Comment · Reviewer_AWMx · 2025-08-05
> >
> > Thank you for your response, I will increase my score.

---

> ### Author Response · Authors · 2025-08-05
> **Thank you for your feedback**
>
> Dear Reviewer AWMx,
>
> Thank you for your review. We appreciate your recognition that the concerns have been addressed. Your review has been very helpful in refining our work.

---

### Note · Authors · 2025-08-12

Dear ACs and Reviewers,

We sincerely thank you for your time and thoughtful feedback, which have greatly contributed to improving our paper. We are encouraged that all reviewers find our definition and motivation of SD-truthfulness both interesting and valuable, and that the proposed Enforced Agreement (EA) mechanism is recognized to be elegant and innovative.

Most of the questions are about the motivations and clarifications of several concepts used in the paper. Below, we provide a brief summary for the AC’s consideration:
* **Sensitivity** (Reviewer WpLs and BNRF). This importance is supported by prior works such as Zhang and Schoenebeck (2023) and Xu et al. (2024). To enhance clarity, we will expand our Appendix with a dedicated discussion explaining why sensitivity is the appropriate concept to focus on, elaborating on our response to Reviewer WpLs.
* **SD-truthfulness for mutual information mechanisms** (Reviewer WpLs). In the current version, we argue that mutual information (MI) mechanisms are unlikely to satisfy SD-truthfulness. Following the reviewer’s suggestion, we will include more detailed justifications by examining three representative MI mechanisms: DMI, pairing MI, and f-MI. As outlined in our rebuttal, it is straightforward to show these mechanisms fail to be SD-truthful.
* **Binary setting** (Reviewer AWMx and X4e2). As clearly stated in the abstract and throughout the paper, the EA mechanism achieves SD-truthfulness specifically in binary-signal settings, while retaining standard truthfulness more generally. We acknowledge this limitation and consider it an interesting direction for future work. Meanwhile, we emphasize that (1) binary-signal settings are widely studied in theory and have practical relevance, and (2) EA offers distinct advantages in terms of simplicity and high sensitivity compared to prior mechanisms.
* **Other minor clarifications**. We will further clarify several potentially confusing concepts—including “information structure,” “self-prediction,” “Bayesian Nash equilibrium,” and “effort”—drawing from our rebuttal discussion. We agree that these improvements will significantly enhance the paper’s accessibility.


Thank you again for your reviews and constructive suggestions.

Best,

Authors of Submission 23815

---

### Decision · Program_Chairs · 2025-09-17

**Decision:**

Accept (poster)

**Comment:**

Traditional peer prediction mechanisms incentivize truthfulness with respect to expected rewards.  In practice, agents may be risk averse or otherwise care about things in a non-linear way.  This paper introduces the idea of ensuring truthfulness stochastically dominates other strategies, which inherently incentivizes a range of utility functions.  An existing rounding approach can achieve this, but makes the scores less sensitive.  A more sophisticated rounding scheme and a new mechanism achieve better sensitivity.

All reviewers are positive about the paper, appreciating the novel, well-motivated definitions, the elegant mechanism designs, and the empirical evaluation.  The main remaining concerns that some reviewers had after the discussion are concerns that the current accessibility of the exposition for readers outside the EC community could be improved (some specific improvements in this regard have been identified and the authors have committed to making them) and that the SD-truthfulness, although not the standard truthfulness, are limited to the binary setting (this limitation receives significant discussion in the paper).